# A circuit mechanism for decision-making biases and NMDA receptor hypofunction

**Sean Edward Cavanagh[1†]\*, Norman H Lam[2†], John D Murray[3‡]\*, Laurence Tudor Hunt[1,4,5,6‡]\*, Steven Wayne Kennerley[1‡]\***

[1]Department of Clinical and Movement Neurosciences, University College London, London, United Kingdom; [2]Department of Physics, Yale University, New Haven, United States; [3]Department of Psychiatry, Yale University School of Medicine, New Haven, United States; [4]Wellcome Trust Centre for Neuroimaging, University College London, London, United Kingdom; [5]Max Planck-UCL Centre for Computational Psychiatry and Aging, University College London, London, United Kingdom; [6]Wellcome Centre for Integrative Neuroimaging, Department of Psychiatry, University of Oxford, Oxford, United Kingdom

**\*For correspondence:**
sean.cavanagh.12@ucl.ac.uk
(SEC);
john.murray@yale.edu (JDM);
laurence.hunt@psych.ox.ac.uk
(LTH);
s.kennerley@ucl.ac.uk (SWK)

[†]These authors contributed
equally to this work
[‡]These authors also contributed
equally to this work

**Competing interests:** The
authors declare that no
competing interests exist.

**Reviewing editor:** Tobias H
Donner, University Medical
Center Hamburg-Eppendorf,
Germany

**Abstract** Decision-making biases can be features of normal behaviour, or deficits underlying neuropsychiatric symptoms. We used behavioural psychophysics, spiking-circuit modelling and pharmacological manipulations to explore decision-making biases during evidence integration. Monkeys showed a pro-variance bias (PVB): a preference to choose options with more variable evidence. The PVB was also present in a spiking circuit model, revealing a potential neural mechanism for this behaviour. To model possible effects of NMDA receptor (NMDA-R) antagonism on this behaviour, we simulated the effects of NMDA-R hypofunction onto either excitatory or inhibitory neurons in the model. These were then tested experimentally using the NMDA-R antagonist ketamine, a pharmacological model of schizophrenia. Ketamine yielded an increase in subjects' PVB, consistent with lowered cortical excitation/inhibition balance from NMDA-R hypofunction predominantly onto excitatory neurons. These results provide a circuit-level mechanism that bridges across explanatory scales, from the synaptic to the behavioural, in neuropsychiatric disorders where decision-making biases are prominent.

## Introduction

A major challenge in computational psychiatry is to relate changes that occur at the synaptic level to the cognitive computations that underlie neuropsychiatric symptoms (*Wang and Krystal, 2014*; *Huys et al., 2016*). For example, one line of research has implicated N-methyl-D-aspartate receptor (NMDA-R) hypofunction in the pathophysiology of schizophrenia (*Nakazawa et al., 2012*; *Kehrer et al., 2008*; *Olney and Farber, 1995*). Some of the strongest evidence in support of this hypothesis comes from the observation that subanaesthetic doses (~0.1–0.5 mg/kg) of the NMDA-R antagonist ketamine produce psychotomimetic effects in humans, especially cognitive aspects (*Krystal et al., 1994*; *Umbricht et al., 2000*; *Malhotra et al., 1996*; see *Frohlich and Van Horn, 2014* for review). But how do we link our understanding of the pharmacological actions of ketamine to its effects on cognition? One strategy to bridge across these different scales is to consider behaviour at the intermediate level of the cortical microcircuit.

Circuit models present a promising avenue to address the challenges of neuropsychiatric research due to their biophysically detailed mechanisms. By perturbing the circuit model at the synaptic level, specific behavioural and neural predictions can be made. For example, NMDA-Rs have long been argued to play a central role in temporally extended cognitive processes such as working memory, supported by studies of the effects of NMDA-R antagonism in prefrontal cortical microcircuits

(*Wang et al., 2013*). Using a cortical circuit model, a precise pattern of working memory deficits can be predicted by hypofunction of NMDA-Rs, eliciting changes in excitation-inhibition balance (E/I ratio) (*Murray et al., 2014*). This predicts changes in behaviour consistent with those observed in healthy volunteers administered with ketamine (*Murray et al., 2014*), and also in patients with schizophrenia (*Starc et al., 2017*). Yet it currently remains unclear whether this approach might generalise to explain the behavioural consequences of NMDA-R antagonism in other temporally extended cognitive processes.

A closely related cognitive process to working memory is evidence accumulation – the decision process whereby multiple samples of information are combined over time to form a categorical choice (*Gold and Shadlen, 2007*). Recent research has advanced our understanding of how such evidence accumulation decisions are made in the healthy brain. Of particular relevance to psychiatric research, it has been possible to disentangle systematic biases in decision-making and reveal the mechanisms through which they occur. For instance, when choosing between two series of bars with distinct heights, people have a preference to choose the option where evidence is more broadly distributed across samples (*Tsetsos et al., 2016*; *Tsetsos et al., 2012*). Although this 'pro-variance bias' may appear irrational, and would not be captured by many normative decision-making models, it becomes the optimal strategy when the accumulation process is contaminated by noise (*Tsetsos et al., 2016*). These behaviours have presently been well-characterised using algorithmic level descriptions of decision formation, yet in order to understand how these decision biases might be affected by NMDA-R hypofunction, a mechanistic explanation is needed.

As with working memory, an influential technique used to investigate evidence accumulation at the mechanistic level has been biophysically grounded computational modelling of cortical circuits (*Wang, 2002*; *Wong and Wang, 2006*; *Murray et al., 2017*). Through strong recurrent connections between similarly tuned pyramidal neurons, and NMDA-R mediated synaptic transmission, these circuits can facilitate the integration of evidence across long timescales. Crucially, these neural circuit models bridge synaptic and behavioural levels of understanding, by predicting both choices and their underlying neural activity. These predictions reproduce key experimental phenomena, mirroring the behavioural and neurophysiological data recorded from macaque monkeys performing evidence accumulation tasks (*Wang, 2002*; *Wong et al., 2007*). Whether neural circuit models can provide a mechanistic implementation of the pro-variance bias, and other systematic biases associated with evidence accumulation, is currently unknown. Moreover, while NMDA-R antagonists have been tested during various decision-making tasks (*Shen et al., 2010*; *Evans et al., 2012*), the role of the NMDA-R in shaping the temporal process of evidence accumulation has not been characterised experimentally.

Here, we used a psychophysical behavioural task in macaque monkeys, in combination with spiking cortical circuit modelling and pharmacological manipulations, to gain new insights into decision-making biases in both health and disease. We trained two subjects to perform a challenging decision-making task requiring the combination of multiple samples of information with distinct magnitudes. Replicating observations from humans, monkeys showed a pro-variance bias. The pro-variance bias was also present in the spiking circuit model, revealing an explanation of how it may arise through neural dynamics. We then investigated the effects of NMDA-R hypofunction in the circuit model, by perturbing NMDA-R function at distinct synaptic sites. Perturbations could either elevate or lower the E/I ratio which strengthen or weaken recurrent circuit dynamics, with each effect making dissociable predictions for evidence accumulation behaviour. These model predictions were tested experimentally by administering monkeys with a subanaesthetic dose of the NMDA-R antagonist ketamine (0.5 mg/kg, intramuscular injection). Ketamine produced decision-making deficits consistent with a lowering of the cortical E/I ratio.

## Results

To study evidence accumulation behaviour in non-human primates, we developed a novel two-alternative perceptual decision-making task (*Figure 1A*). Subjects were presented with two series of eight bars (evidence samples), one on either side of central fixation. Their task was to decide which evidence stream had the taller/shorter average bar height, and indicate their choice contingent on a contextual cue shown at the start of the trial. The individual evidence samples were drawn from Gaussian distributions, which could have different variances for different options (*Figure 1B*). This

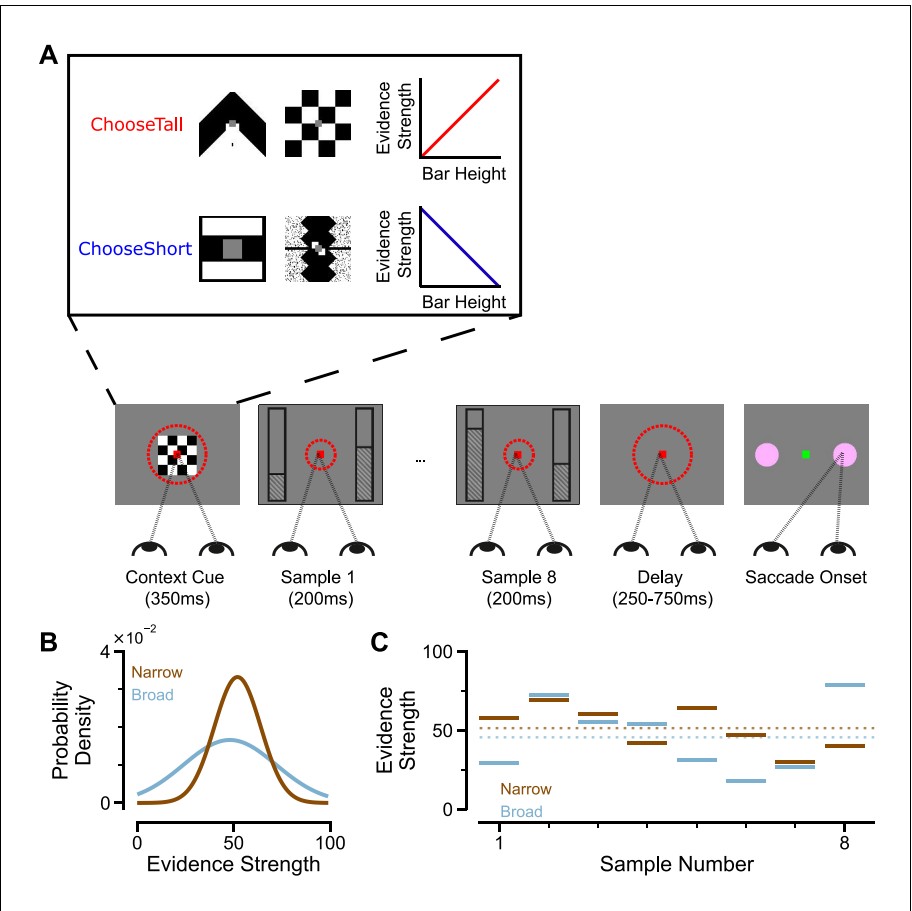

**Figure 1.** An evidence-varying decision-making task for macaque monkeys. (**A**) Task design. Two streams of stimuli were presented to a monkey, both of which consisted of a sequence of eight samples of bars of varying heights. Depending on the contextual cue shown at the start of the trial, the monkey had to report the stream with either taller or shorter mean bar height. On correct trials, the monkey was rewarded proportionally to the mean evidence for the correct stream; incorrect trials were not rewarded. The monkey was required to fixate centrally while the evidence was presented, indicated by the dashed red fixation zone (not visible to subject). (**B**) Generating process of each stimulus stream. The generating mean for each trial was chosen from a uniform distribution (see Materials and methods), while the generating standard deviation was 12 and 24 for the narrow (brown) and broad (blue) streams respectively. (**C**) Example Trial. The bar heights in both streams varied over time. The dotted lines illustrate the mean of the eight stimuli for the narrow/broad streams. In this example, the narrow stream has taller mean bar height and thus more mean evidence strength, so is the correct choice. The narrow/broad streams are randomly assigned to the left/right options on different trials; in the example trial shown here (A and C), the narrow stream is assigned to the right option, the broad stream is assigned to the left option.

task design had several advantages over evidence accumulation paradigms previously employed with animal subjects. Subjects were given eight evidence samples with distinct magnitudes (*Figure 1C*) – encouraging a temporal integration decision-making strategy. Precise experimental control of the stimuli facilitated analytical approaches probing the influence of evidence variability and time course on choice, and allowed us to design specific trials that attempted to induce systematic biases in choice behaviour.

Two monkeys (*Macaca mulatta*) completed 29,726 trials (Monkey A: 10,748; Monkey H: 18,978). Despite the challenging nature of the task, subjects were able to perform it with high accuracy (*Figure 2A–B*, *Figure 2—figure supplement 1A–B*). The precise control of the discrete stimuli allowed us to evaluate the impact of evidence presented at each time point on the final behavioural choice, via logistic regression (see Materials and methods). Stimuli presented at a time point with a larger regression coefficient have a strong impact on the choice, relative to time points with smaller

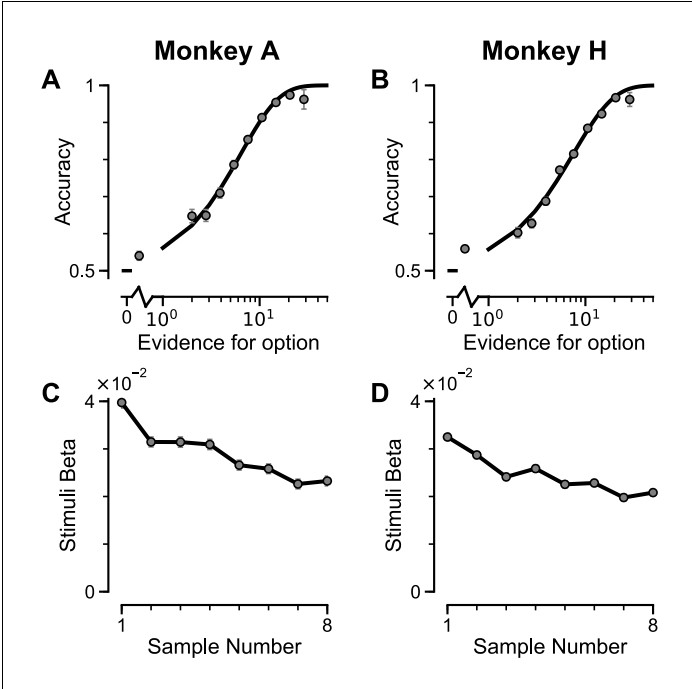

**Figure 2.** Subjects use evidence presented throughout the trial to guide their choices. (A-B) Choice accuracy plotted as a function of the amount of evidence in favour of the best option. Lines are a psychometric fit to the data. (C-D) Logistic regression coefficients reveal the contribution (weight) of all eight stimuli on subjects' choices (see Materials and methods). Although subjects used all eight stimuli to guide their choices, they weighed the initially presented evidence more strongly. All errorbars indicate the standard error.

The online version of this article includes the following figure supplement(s) for figure 2:

**Figure supplement 1.** Subjects use evidence presented throughout the trial to guide their choices – data separated by 'ChooseTall' and 'ChooseShort' trials.

coefficients. We found that the subjects utilised all eight stimuli throughout the trial to inform their decision, and demonstrated a primacy bias such that early stimuli have stronger temporal weights than later stimuli (*Figure 2C–D*, *Figure 2—figure supplement 1C–D*). A primacy bias has been reported in prior studies in monkeys, and is consistent with a decision-making strategy of bounded evidence integration (*Kiani et al., 2008*; *Nienborg and Cumming, 2009*; *Wimmer et al., 2015*). As it was clear both monkeys could accurately perform the task, all subsequent figures are presented with data collapsed across subjects for conciseness, but results separated by subjects are consistent (see supplementary figures).

We next probed the influence of evidence variability on choice. We designed specific choice options with different levels of standard deviation across samples in an attempt to replicate the pro-variance bias previously reported for human subjects (see Materials and methods) (*Tsetsos et al., 2016*; *Tsetsos et al., 2012*). On each trial, one option was allocated a narrow distribution of bar heights, and the other a broad distribution. In different conditions, either the broad or narrow stimuli stream could be the correct choice (*'Broad Correct' Trials* or *'Narrow Correct' Trials*), or there could be no clear correct answer (*'Ambiguous' Trials*) (*Figure 3A*, *Figure 3—figure supplements 1* and *2*). If subjects chose optimally, and only the mean bar height influenced their choice, their accuracy would be the same in *'Broad Correct'* and *'Narrow Correct'* trials and they would be indifferent to the variance of the distributions in *'Ambiguous'* trials. We show that our monkeys deviate from such behaviours. The monkeys are more accurate on *'Broad Correct'* trials than on *'Narrow Correct'* trials (*Figure 3B*, *Figure 3—figure supplements 1* and *2*). Furthermore, in the *'Ambiguous'* trials, the monkeys demonstrated a preference for the broadly distributed stream, which has greater variability across samples (*Figure 3C*, *Figure 3—figure supplements 1* and *2*). Such a pro-variance bias

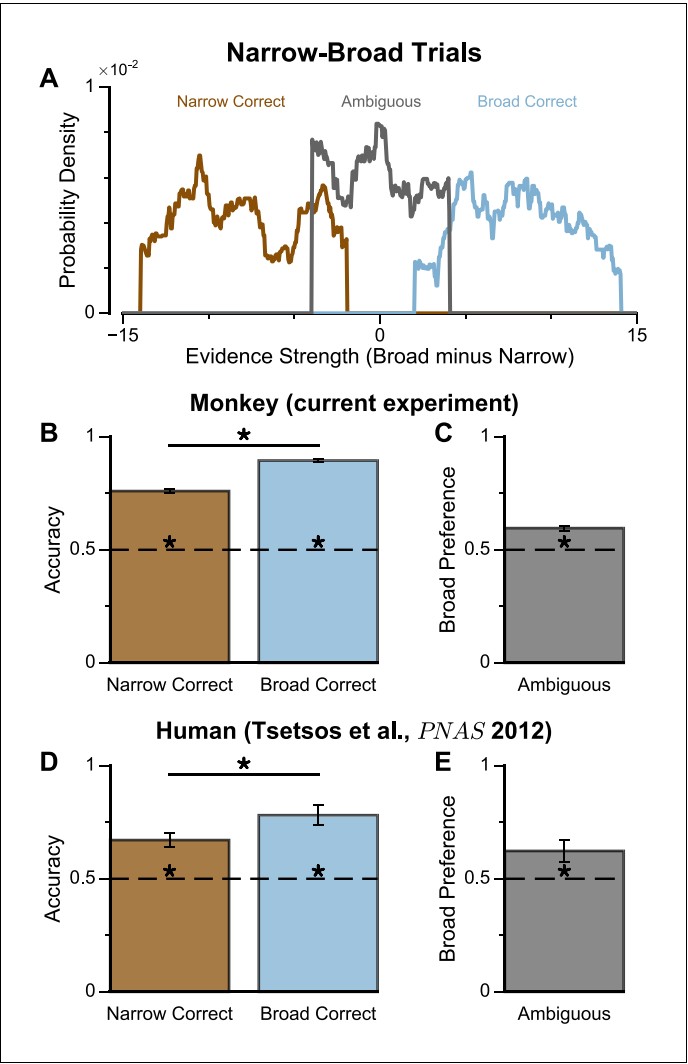

**Figure 3.** Subjects show a pro-variance bias in their choices on Narrow-Broad Trials, mirroring previous findings in human subjects. (**A**) The narrow-broad trials include three types of conditions, where either the narrow stream is correct (brown), the broad stream is correct (blue), or the difference in mean evidence is small (grey, 'Ambiguous' trials). See Materials and methods and *Figure 3—figure supplement 1* for details of the generating process. (**B**–**C**) Monkey choice performance on Narrow-Broad trials. (**B**) Subjects were significantly more accurate on 'Broad-correct' trials (Chi-squared test, chi = 99.05, $p < 1 \times 10^{-10}$). Errorbars indicate the standard error. (**C**) Preference for the broad option on 'Ambiguous' trials. Subjects were significantly more likely to choose the broad option (Binomial test, $p < 1 \times 10^{-10}$). Errorbar indicates the standard error. (**D**–**E**) Human choice performance on Narrow-Broad trials previously reported by *Tsetsos et al., 2012*. (**D**) Choice accuracy when either the narrow or the broad stream is correct, respectively. Subjects were more accurate on 'Broad-correct' trials. (**E**) Preference for the broad option on 'Ambiguous' trials. Subjects were more likely to choose the broad option.

The online version of this article includes the following figure supplement(s) for figure 3:

**Figure supplement 1.** Extra Information on Narrow-Broad Trials, separated by subjects.
**Figure supplement 2.** Extra Information on Narrow-Broad Trials, separated by 'ChooseTall' and 'ChooseShort' trials.

pattern of decision behaviour is similar to what was found in human subjects (*Tsetsos et al., 2016*; *Tsetsos et al., 2012*; *Figure 3D–E*).

To further probe the pro-variance bias, we studied choices from a larger pool of 'Regular' trials in which the mean evidences and variabilities of the two streams were set independently on each trial (*Figure 4A–B*, *Figure 4—figure supplements 1* and *2*). 'Regular' trials allowed us to explore the

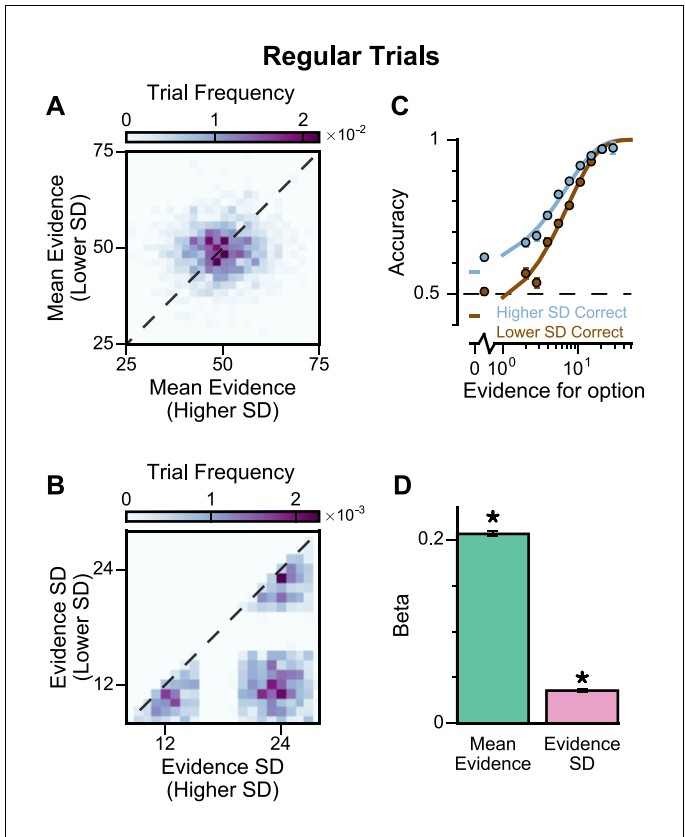

**Figure 4.** Subjects show a pro-variance bias in their choices on regular trials. For these analyses, stimulus streams were divided into 'Lower SD' or 'Higher SD' options post-hoc, on a trial-wise basis. (**A**) On regular trials, the mean evidence of each stream was independent. (**B**) Each stream is sampled from either a narrow or a broad distribution, such that about 50% of the trials have one broad stream and one narrow stream, 25% of the trials have two broad streams, and 25% of the trials have two narrow streams. (**C**) Psychometric function when either the 'Lower SD' (brown) or 'Higher SD' (blue) stream is correct in the regular trials. (**D**) Regression analysis using the left-right differences of the mean and standard deviation of the stimuli evidence to predict left choice. The beta coefficients quantify the contribution of both statistics to the decision-making processes of the monkeys (Mean Evidence: t = 74.78, p<$10^{-10}$; Evidence Standard Deviation: t = 19.65, p<$10^{-10}$). Notably, a significantly positive evidence SD coefficient indicates the subjects preferred to choose options which were more variable across samples. Errorbars indicate the standard error.

The online version of this article includes the following figure supplement(s) for figure 4:

**Figure supplement 1.** Extra information on Regular Trials, separated by subjects.

**Figure supplement 2.** Extra information on Regular Trials, separated by 'ChooseTall' and 'ChooseShort' trials.

**Figure supplement 3.** Extra information on Regular Trials – the subjects do not show a frequent winner bias.

pro-variance bias across a greater range of choice difficulties (**Figure 4C**) and quantitatively characterise its effect using regression analysis. On 'Regular' trials, subjects also demonstrated a preference for options with broadly distributed evidence. Regression analysis confirmed that evidence variability was a significant predictor of choice (**Figure 4D**; see Materials and methods).

In addition, we defined the pro-variance bias (PVB) index as the ratio of the regression coefficient for evidence standard deviation over the regression coefficient for mean evidence. Although evidence standard deviation was irrelevant in determining the correct option to choose in the task, and it is important to note that we were not suggesting it is explicitly computed by the monkey subjects, the sensitivity of choice behaviour to evidence standard deviation could also arise as a by-product of the neural computations to evaluate the task-relevant mean evidence (as shown later in the Results). PVB index thus served as a unitless, descriptive measure of the evidence accumulation process, quantifying the subjects' sensitivity to evidence standard deviation relative to the evidence

accumulation process. A PVB index value of 0 thereby indicates no pro-variance bias, whereas a PVB index value of 1 indicates the subject is as sensitive to evidence standard deviation as they are to mean evidence. The PVB index thus provides a quantitative measure of the pro-variance bias. A key motivation for defining PVB index is as a potentially useful measure for assessing changes in decision-making behaviour, such as via pharmacological perturbation (performed in later experiments in this paper). For example, if a perturbation simply weakened the overall sensitivity of choice to stimulus information, this would presumably down-scale the mean and standard deviation regression coefficients proportionally, yielding no change in the PVB index as a ratio. In contrast, if a perturbation to the decision-making process differentially impacts how evidence mean vs. standard deviation impact choice, then this would be reflected as a change in the PVB index. From the 'Regular' trials, the PVB index across both monkeys was 0.173 (Monkey A = 0.230; Monkey H = 0.138).

Recent work has suggested that when traditional evidence accumulation tasks are performed, it is hard to dissociate whether subjects are combining information across samples, or whether conventional analyses may be disguising a simpler heuristic (*Waskom and Kiani, 2018*; *Stine et al., 2020*). In particular, an alternative decision-making strategy which does not involve temporal accumulation of evidence is to detect the single most extreme sample. Because the extreme sample will occur at different times in each trial, if a subject employed this strategy, the choice regression weights across time points would be distributed as in *Figure 2C–D*. Therefore, it is possible for these findings to be mistakenly interpreted as reflecting evidence accumulation. We wanted to quantitatively confirm that subjects were using the strategy we envisioned when designing our task, namely evidence accumulation. Additionally, we wanted to further investigate the relative contributions of mean evidence and evidence variability on choices. A logistic regression approach probed the influence upon choice of mean evidence, evidence variability, first/last samples, and the most extreme samples within each stream (*Figure 4—figure supplements 1E,H* and *2C,F,I,L*, see Materials and methods). A cross-validation approach revealed choice was principally driven by the mean evidence, verifying that subjects performed the task using evidence accumulation (*Supplementary file 1*, see Materials and methods).

Although this analysis revealed choices were not primarily driven by an 'extreme sample detection' decision strategy, another concern was whether partially employing this strategy could explain the pro-variance effect we observed. To address this, we compared the influence of 'evidence variability' versus the influence of 'extreme samples' on subjects' choices. Cross-validation revealed that choices were better described by a model incorporating evidence variability, rather than the extreme sample values (*Supplementary file 2*). We also demonstrated that including evidence variability as a co-regressor improved the performance of all combinations of nested models (*Supplementary file 3*). In summary, it can be concluded that although subjects integrated across samples, they were additionally influenced by sample variability.

Previous studies have revealed that a 'frequent winner' bias – whereby subjects prefer to choose options where there is a greater number of cases of stronger evidence between the simultaneously presented stimuli – coexists with the pro-variance bias (*Tsetsos et al., 2016*). Furthermore, both of these biases may arise from the same selective accumulation mechanism (*Tsetsos et al., 2016*). Therefore, we next analysed whether our subjects' choices were also influenced by a 'frequent winner' bias (*Figure 4—figure supplement 3*; Materials and methods). After controlling for the influence of mean evidence on choices, we found that neither subject demonstrated a 'frequent winner' bias.

Existing algorithmic-level proposals for generating a pro-variance bias in human decision-making rely on the disregarding of sensory information before it enters the accumulation process, depending on its salience (*Tsetsos et al., 2016*). To investigate a possible alternative basis for the pro-variance bias, at the level of neural implementation, we sought to characterise decision-making behaviour in a biophysically-plausible spiking cortical circuit model (*Figure 5A–B*, *Figure 5—figure supplement 1*; *Wang, 2002*; *Lam, 2017*). In the circuit architecture, two groups of excitatory pyramidal neurons are assigned to the left and right options, such that high activity in one group signals the response to the respective option. Excitatory neurons within each group are recurrently connected to each other via AMPA and NMDA receptors, and this recurrent excitation supports ramping activity and evidence accumulation. Both groups of excitatory neurons are jointly connected to a group of inhibitory interneurons, resulting in feedback inhibition and winner-take-all competition (*Wang, 2002*; *Wong and Wang, 2006*). The two groups of excitatory neurons receive separate

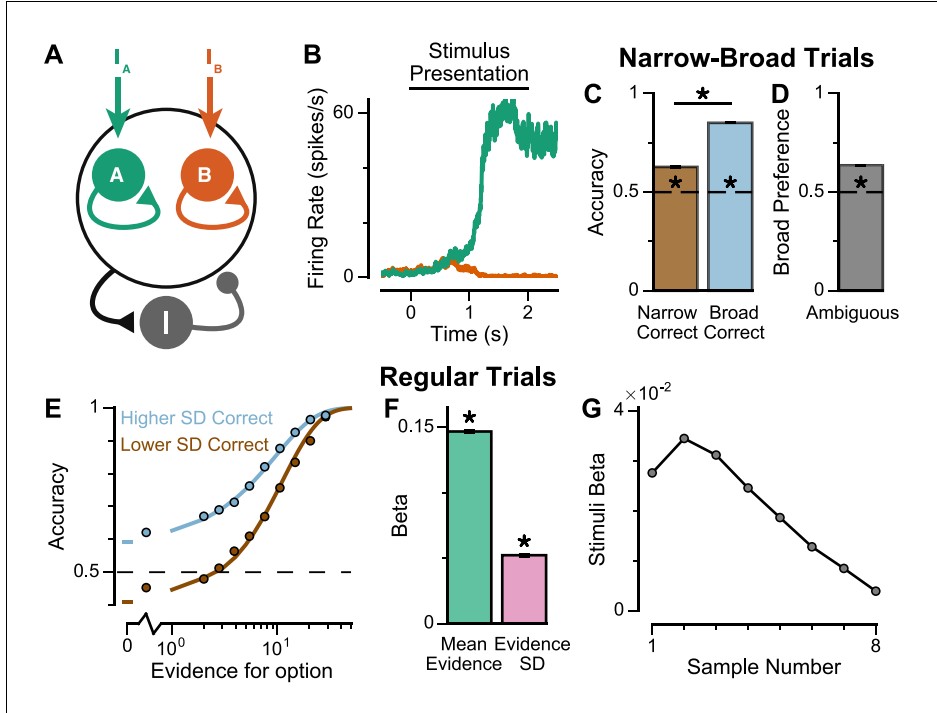

**Figure 5.** Spiking cortical circuit model reproduces pro-variance bias. (A) Circuit model schematic. The model consists of two excitatory neural populations which receive separate inputs ($I_A$ and $I_B$), each reflecting the momentary evidence for one of the two stimuli streams. Each population integrates evidence due to recurrent excitation, and competes with the other via lateral inhibition mediated by a population of interneurons. (B) Example firing rate trajectories of the two populations on a single trial where option A is chosen. (C, D) Narrow-Broad Trials. (C) The circuit model is significantly more accurate when the broad stream is correct, than when the narrow stream is correct (Chi-squared test, chi = 1981, $p<1\times10^{-10}$). (D) On 'Ambiguous trials', the circuit model is significantly more likely to choose the broad option (Binomial test, $p<1\times10^{-10}$). (E–G) Regular trials. (E) The psychometric function of the circuit model when either the 'Lower SD' (brown) or 'Higher SD' (blue) stream is correct, respectively. (F) Regression analysis of the circuit model choices on regular trials, using evidence mean and variability as predictors of choice. Both quantities contribute to the decision-making process of the circuit model (Mean Evidence: t = 129.50, $p<10^{-10}$; Evidence Standard Deviation: t = 45.27, $p<10^{-10}$). (G) Regression coefficients of the stimuli at different time-steps, showing the time course of evidence integration. The circuit demonstrates a temporal profile which decays over time, similar to the monkeys. All errorbars indicate the standard error.

The online version of this article includes the following figure supplement(s) for figure 5:

**Figure supplement 1.** Extended regression results on the circuit model performance.

**Figure supplement 2.** Pro-variance bias and temporal weightings in trials separated with more or less total evidence, for circuit model and monkey data.

---

inputs - with each group receiving information about one of the two options (i.e. Group A receives $I_A$ reflecting the left option; Group B receives $I_B$ reflecting the right option). Specifically, we assume the bar heights from each stream are remapped, upstream of the simulated decision-making circuit, to evidence for the corresponding option depending on the cued context. Therefore, taller bars correspond to larger inputs in a *'ChooseTall'* trial and smaller inputs in a *'ChooseShort'* trial. Combined together, this synaptic architecture endows the circuit model with decision-making functions.

The spiking circuit model was tested with the same trial types as the monkey experiment. Importantly, not only can the circuit model perform the evidence accumulation task, it also demonstrated a pro-variance bias comparable to the monkeys (*Figure 5C–F*). Regression analysis showed that the circuit model utilises a strategy similar to the monkeys to solve the decision-making task (*Figure 5— figure supplement 1B*). The temporal process of evidence integration in the circuit model disproportionately weighted early stimuli over late stimuli (*Figure 5G*), similar to the evidence integration

patterns observed in both monkeys. However, the circuit model demonstrated an initial ramp-up in stimuli weights due to the time needed for it to reach an integrative state.

To understand the origin of the pro-variance bias in the spiking circuit, we mathematically reduced the circuit model to a mean-field model (*Figure 6A*), which demonstrated similar decision-making behaviour to the spiking circuit (*Figure 6B-C*, *Figure 6—figure supplement 1*). The mean-field model, with two variables representing the integrated evidence for the two choices, allowed phase-plane analysis to further investigate the pro-variance bias. A simplified case was considered where the broad and narrow streams have the same mean evidence, and the stimuli evidence varies over time in the broad stream but not the narrow stream (i.e. $\sigma_N = 0$) (*Figure 6E-H*). This example provides an intuitive explanation for the pro-variance bias: a momentarily strong stimulus has an asymmetrically greater influence upon the decision-making process than a momentarily weak stimulus. Input streams with larger variability and thus a higher chance to display both strong inputs and weak inputs, can thus leverage such asymmetry more so than input streams with small variability, resulting in pro-variance bias. Such asymmetry arises from the expansive non-linearities of the firing rate profiles (*Figure 6D*, see Materials and methods).

To explore whether this explanation may account for the pro-variance bias in the circuit model and monkey behaviour (*Figures 4* and *5*), we re-analysed the data separating trials into two halves: those with more or less total evidence (summed across both streams) (*Figure 5—figure supplement*

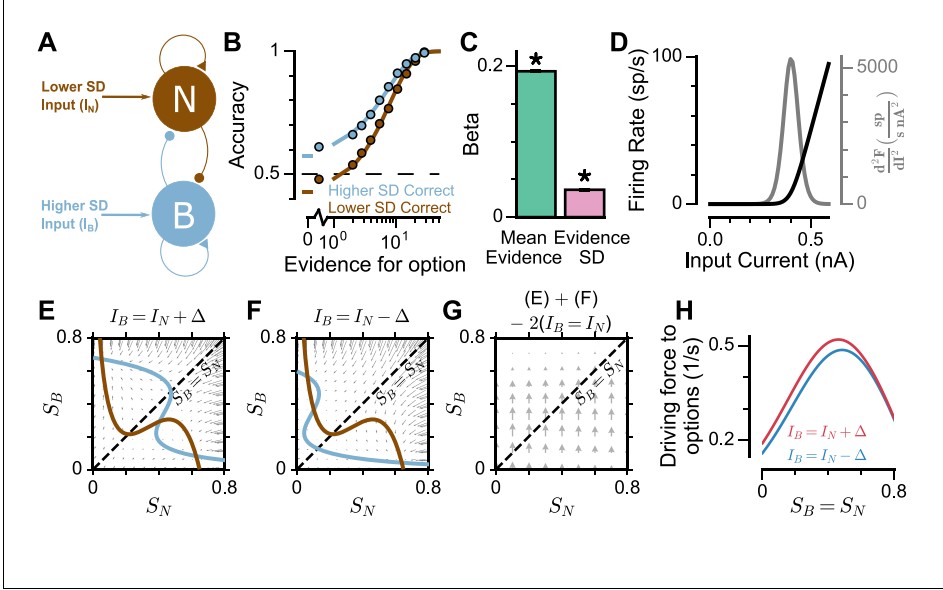

**Figure 6.** Mean-Field model explanation for pro-variance bias. (A) The mean-field model of the circuit, with two variables representing evidence for the two options. For simplicity, we assume one stream is narrow and one is broad, and label the populations receiving the inputs as N and B respectively. (B) Psychometric function of regular trials as in (*Figure 5E*). (C) Regression analysis of the regular trial data as in (*Figure 5F*) (mean: t = 143.42, p < $10^{-10}$; standard deviation: t = 30.76, p < $10^{-10}$). Errorbars indicate the standard error. (D) The mean-field model uses a generic firing rate profile (black), with zero firing rate at small inputs, then a near-linear response as input increases (see Materials and methods). Such profiles have an expansive non-linearity (with a positive second order derivative (grey)) that can generate pro-variance bias. (E–H) An explanation of the pro-variance bias using phase-plane analysis. (E) A momentarily strong stimulus from the broad stream will drive the model to choose broad (large $S_B$, small $S_N$). Blue and brown lines correspond to nullclines. (F) A momentarily weak stimulus in the broad stream will drive the model to choose narrow (large $S_N$, small $S_B$). (G) The net effect of one strong and one weak broad stimulus, compared with two average stimuli, is to drive the system to the broad choice. That is, a momentarily strong stimulus has an asymmetrically greater influence on the decision-making process than a momentarily weak stimulus, leading to pro-variance bias. (H) The net drive to the broad or narrow option when the broad stimulus is momentarily strong (red) or weak (blue), along the diagonal ($S_B = S_N$ in G).

The online version of this article includes the following figure supplement(s) for figure 6:

**Figure supplement 1.** Extended regression results on the mean-field model performance.

*2*). The circuit model demonstrated a smaller PVB index (larger mean evidence and smaller evidence standard deviation regression weights) for trials with more total evidence than for trials with less total evidence (*Figure 5—figure supplement 2C*). This was consistent with the prediction from the F-I non-linearity: trials with more total evidence, and thus larger total input, will more strongly drive the neurons to the near-linear regime of the firing rate profile, where the effect of the expansive non-linearity was weaker (*Figure 6D*). Similar analysis of the monkey behavioural data revealed a similar trend of smaller PVB index (larger mean evidence and smaller evidence standard deviation regression weights) for trials with more total evidence than less total evidence, though the effect was insignificant (*Figure 5—figure supplement 2G*). In addition, distinct temporal weighting on stimuli were observed in both the circuit model and experimental data, for trials with more versus less total evidence (*Figure 5—figure supplement 2D,H*).

An advantage of the circuit model over existing algorithmic level explanations of the pro-variance bias is it can be used to make testable behavioural predictions in response to different synaptic or cellular perturbations, including excitation-inhibition (E/I) imbalance. In turn, perturbation experiments can constrain and refine model components. Therefore, we studied the behavioural effects of distinct E/I perturbations, and upstream sensory deficit, on decision making and in particular, pro-variance bias (*Figure 7*, *Figure 7—figure supplement 1*). Three perturbations were introduced to the circuit model: lowered E/I balance (via NMDA-R hypofunction on excitatory pyramidal neurons), elevated E/I balance (via NMDA-R hypofunction on inhibitory interneurons), or sensory deficit (as weakened scaling of external inputs to stimuli evidence) (*Figure 7A*).

While all circuit models were capable of performing the task (*Figure 7B–E*), the choice accuracy of each perturbed model was reduced when compared to the control model. This was quantified by the regression coefficient of mean evidence (*Figure 7F*). In addition, the regression coefficient for evidence standard deviation was reduced for each perturbed model in comparison to the control model, indicating a lesser influence of evidence variability on choice (*Figure 7G*). Finally, in a dissociation between the three model perturbations, the PVB index was increased by lowered E/I, decreased by elevated E/I, and roughly unaltered by sensory deficits (*Figure 7H*). Further regression analyses indicated no obvious shift in utilised strategies relative to the control model (*Figure 7—figure supplement 1*). Crucially, the effect of E/I and sensory perturbations on PVB index and regression coefficients were generally robust to the strength and pathway of perturbation (*Figure 7—figure supplements 2* and *3*).

Disease and pharmacology-related perturbations likely concurrently alter multiple sites, for instance NMDA-Rs of both excitatory and inhibitory neurons. We thus parametrically induced NMDA-R hypofunction on both excitatory and inhibitory neurons in the circuit model. The net effect on E/I ratio depended on the relative perturbation strength to the two populations (*Lam, 2017*). Stronger NMDA-R hypofunction on excitatory neurons lowered the E/I ratio, while stronger NMDA-R hypofunction on inhibitory neurons elevated the E/I ratio. Notably, proportional reduction to both pathways preserved E/I balance and did not lower the mean evidence regression coefficient (a proxy of performance) (*Figure 7—figure supplement 2A*). On the other hand, decision making performance was maximally susceptible to perturbations in the orthogonal direction, along the E/I axis (*Lam, 2017*). Furthermore, along this axis PVB index monotonically increased with lowered E/I ratio and decreased with elevated E/I ratio, demonstrating a robust prediction from our circuit model (*Figure 7—figure supplement 2C*). Sensory deficit perturbations did not significantly alter PVB index, until the limit where decision making performance was greatly impaired (*Figure 7—figure supplement 4*). Finally, the temporal weightings were distinctly altered by the elevated- and lowered- E/I perturbations (*Figure 7I*). The circuit model thus provided the basis of dissociable prediction by E/I-balance perturbing pharmacological agents.

Since the decision making choice accuracy depends on E/I ratio along an inverted-U shape – where the control, E/I balanced model is right next to the (slightly lowered E/I) peak (*Lam, 2017*)- both elevating and lowering E/I ratio drive the model away from the peak, resulting in lowered mean evidence regression weight. The evidence standard deviation regression weight similarly follows an inverted-U shape, but with the peak at a more strongly lowered E/I ratio (*Figure 7—figure supplement 2*). As such, elevating E/I ratio consistently lowers the evidence standard deviation regression weight, but lowering E/I ratio by a small amount initially increases the evidence standard deviation regression weight, and only decreases the evidence standard deviation regression weight after such peak in the inverted-U shape is passed at greater perturbation strengths. Notably,

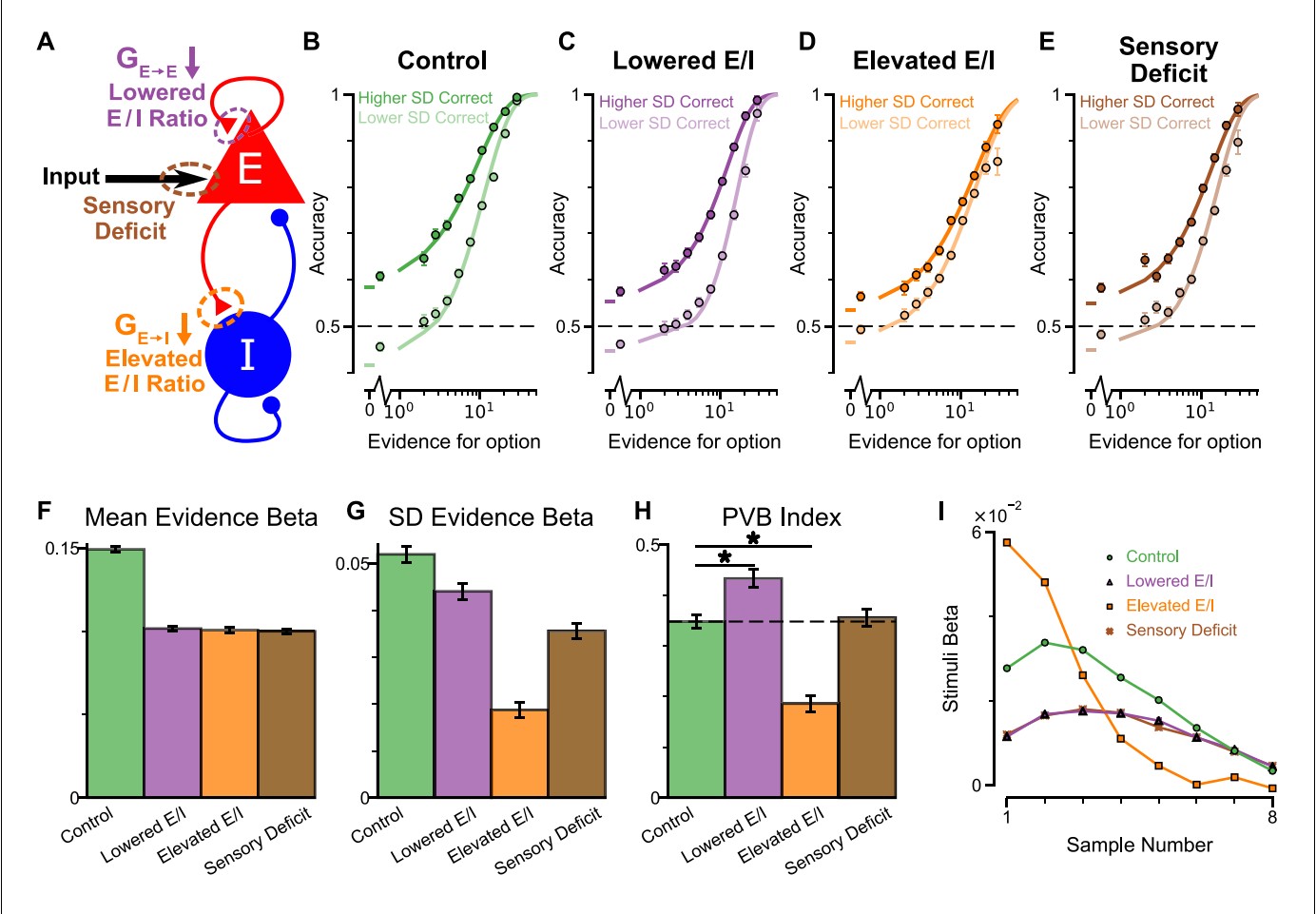

**Figure 7.** Predictions for E/I perturbations of the Spiking Circuit Model. (A) Model perturbation schematic. Three potential perturbations are considered: lowered E/I (via NMDA-R hypofunction on excitatory pyramidal neurons), elevated E/I (via NMDA-R hypofunction on inhibitory interneurons), or sensory deficit (as weakened scaling of external inputs to stimuli evidence).(B–E) The regular-trial choice accuracy for each of the circuit perturbations (dark colour for when the 'Higher SD' stream is correct, light colour for when the 'Lower SD' stream is correct).(F–H) Regression analysis on the regular trial choices of the four models, using evidence mean and evidence variability to predict choice.(F) The mean evidence regression coefficients in the four models. Lowering E/I, elevating E/I, and inducing sensory deficits similarly reduce the coefficient, reflecting a drop in choice accuracy. (G) The evidence standard deviation regression coefficients in the four models. All three perturbations reduce the coefficient, but to a different extent. (H) The PVB index (ratio of evidence standard deviation coefficient over mean evidence coefficient) provides dissociable predictions for the perturbations. The lowered E/I circuit increases the PVB index relative to the control model (permutation test, $p=6\times10^{-5}$), while the elevated E/I circuit decreases the PVB index (permutation test, $p<10^{-5}$). The PVB index is roughly maintained in the sensory deficit circuit (permutation test, $p=0.6933$). The dashed line indicates the PVB index for the control circuit, * indicates significant difference when the PVB index is compared with the control circuit. (I) The regression weights of stimuli at different time-steps for the four models. All errorbars indicate the standard error.

The online version of this article includes the following figure supplement(s) for figure 7:

**Figure supplement 1.** Model perturbations do not influence decision-making strategy.

**Figure supplement 2.** Regression analysis using evidence mean and evidence variability to predict choice, under simultaneous NMDA-R hypofunctions on excitatory and inhibitory neurons.

**Figure supplement 3.** Regression analysis using mean, maximum, minimum, first, and last evidence values of each of the left and right streams as regressors, under simultaneous NMDA-R hypofunctions on excitatory and inhibitory neurons.

**Figure supplement 4.** Regression coefficients and PVB index as a function of sensory deficit.

regardless of the magnitude with which E/I ratio is lowered, PVB index is consistently increased, providing a robust measure of pro-variance bias.

To explore these predictions experimentally, we collected behavioural data from both monkeys following the administration of a subanaesthetic dose (0.5 mg/kg, intramuscular injection) of the NMDA-R antagonist ketamine (see Materials and methods, *Figure 8*, *Figure 8—figure supplement*

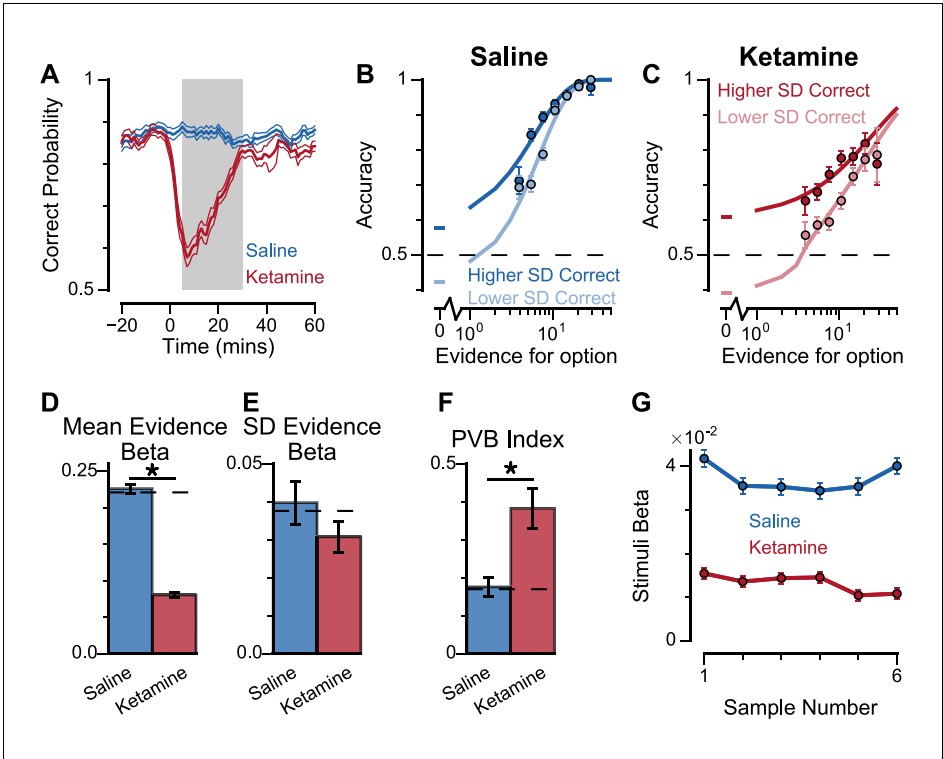

**Figure 8.** Experimental effects of ketamine on evidence accumulation behaviour produce an increased pro-variance bias, consistent with lowered excitation-inhibition balance. (**A**) Mean percentage of correct choices across sessions made by monkeys relative to the injection of ketamine (red) or saline (blue). Shaded region denotes 'on-drug' trials (trials 5–30 min after injection) which are used for analysis in the rest of the figure. (**B, C**) The psychometric function when either the 'Lower SD' or 'Higher SD' streams are correct, with saline (**B**) or ketamine (**C**) injection. (**D–F**) Ketamine injection impairs the decision-making of the monkeys, in a manner consistent with the prediction of the lowered E/I circuit model. Dashed lines indicate pre-injection values in each plot. (**D**) The regression coefficient for mean evidence, under injection of saline or ketamine. Ketamine significantly reduces the coefficient (permutation test, $p < 1 \times 10^{-6}$), reflecting a drop in choice accuracy. (**E**) The evidence standard deviation regression coefficient, under injection of saline or ketamine. Ketamine does not significantly reduce the coefficient (permutation test, $p = 0.152$). (**F**) Ketamine increases the PVB index (permutation test, $p = 8 \times 10^{-6}$), consistent with the model prediction of the lowered E/I circuit. (**G**) The regression weights of stimuli at different time-steps, for the monkeys with saline or ketamine injection. Ketamine injection lowers and flattens the curve of temporal weights, consistent with the lowered E/I circuit model. Errorbars in (**A**) indicate the standard error mean, in all other panels errorbars indicate the standard error.

The online version of this article includes the following figure supplement(s) for figure 8:

**Figure supplement 1.** Extra information on ketamine experiments, separated by subjects.

**Figure supplement 2.** Behavioural effects of ketamine on the pro-variance bias and temporal weightings are not explained by lapsing.

**Figure supplement 3.** Time course of ketamine's influence on pro-variance bias.

**Figure supplement 4.** Cosine similarity of various perturbation effects in the circuit model, to the effect of ketamine injections on monkey behaviour.

**Figure supplement 5.** Euclidean distance of various perturbation effects in the circuit model, to the effect of ketamine injections on monkey behaviour.

**Figure supplement 6.** Kullback–Leibler (KL) divergence from monkey behavior with saline or ketamine to circuit models with various perturbations.

**Figure supplement 7.** Psychometric function of monkeys under ketamine injection, and circuit model with large sensory deficit.

**Figure supplement 8.** Predictions for E/I perturbations of the Spiking Circuit Model, with built-in Monkey A lapse rate, compared with Monkey A behaviour.

**Figure supplement 9.** Predictions for E/I perturbations of the Spiking Circuit Model, with built-in Monkey H lapse rate, compared with Monkey H behaviour.

1). After a baseline period of the subjects performing the task, either ketamine or saline was injected intramuscularly (Monkey A: 13 saline sessions, 15 ketamine sessions; Monkey H: 17 saline sessions, 18 ketamine sessions). Administering ketamine had behavioural effects for around 30 min in both subjects. The data collected during this period formed a behavioural database of 8521 completed trials (Monkey A Saline: 1710; Monkey A Ketamine: 2276; Monkey H Saline: 2669; Monkey H Ketamine: 1866). Following ketamine administration, subjects' choice accuracy was markedly decreased (*Figure 8A*), without a significant shift in their strategies (*Figure 8—figure supplement 1*, *Supplementary file 4*).

To understand the nature of this deficit, we studied the effect of drug administration on the pro-variance bias (*Figure 8B–F*). Although subjects were less accurate following ketamine injection, they retained a pro-variance bias (*Figure 8C*). Regression analysis confirmed ketamine caused choices to be substantially less driven by mean evidence (*Figure 8D*), but still strongly influenced by the standard deviation of evidence across samples (*Figure 8E*). The PVB index was significantly higher when ketamine was administered, than saline (permutation test p=8×10$^{-6}$, *Figure 8F*). Of all the circuit model perturbations, this was only consistent with lowered E/I balance (*Figure 7H*).

In further analysis, we also controlled for the influence of ketamine on the subjects' lapse rate – i.e. the propensity for the animals to respond randomly regardless of trial difficulty. We modelled this lapse rate using an additional term that bounded the logistic function at $Y_0$ and (1-$Y_0$), rather than 0 and 1 (*Figure 8—figure supplement 2*, see Materials and methods, *Equation 9*). In other words, lapse rate refers to the asymptote error rate at the limit of strong evidence. Consistent with the psychometric function (*Figure 8C*), we found that ketamine significantly increased the subjects' lapse rate (Subject A: Lapse$_{(Saline)}$=1.49×10$^{-11}$, Lapse$_{(Ketamine)}$=0.118, permutation test, p<0.0001; Subject H: Lapse$_{(Saline)}$=0.012, Lapse$_{(Ketamine)}$=0.0684, permutation test, p=0.019). Crucially, however, the PVB effect was still present in the regression model that included the effect of lapses. This confirms that the change in lapse rate was not responsible for any of the behavioural effects of ketamine outlined above. We also investigated the time-course of ketamine's influence on the PVB index (*Figure 8—figure supplement 3*). This confirmed that the rise in PVB was an accurate description of a common behavioural deficit throughout the duration of ketamine administration.

Additional observations further supported the lowered E/I hypothesis for the effect of ketamine on monkey choice behaviour. Quantitative model comparison, using cosine similarity, Euclidean distance, and Kullback–Leibler (KL) divergence, revealed the effect of ketamine injection on monkey choice behaviour was better explained by lowered E/I perturbations in the circuit model, than by sensory deficit or elevated E/I perturbations (*Figure 8—figure supplements 4–6*). Very strong sensory deficit may also increase PVB index, but with minimal decision making performance and a psychometric function very different from the monkey data (*Figure 7—figure supplement 4*, *Figure 8—figure supplement 7*). In addition, we investigated the effect of ketamine on the time course of evidence weighting (*Figure 8G*). It caused a general downward shift of the temporal weights; but had no strong effects on how each stimulus was weighted relative to the others in the stream. This shifting of the weights could reflect a sensory deficit, but given the results of the pro-variance analysis, collectively the behavioural effects of ketamine are most consistent with lowered E/I balance and weakened recurrent connections. Notably, the saline data demonstrate a U-shaped pattern different from the primacy pattern observed in non-drug experiments (*Figure 2C,D*) and spiking circuit models (*Figure 7I*). This may be due to task modifications for the ketamine/saline experiments compared with the non-drug experiments, but could also potentially arise from distinct regimes of decision making attractor dynamics (e.g. see *Genís Prat-Ortega et al., 2020*).

To quantify the effect of lapse rate on evidence sensitivity and regression weights in general, we examined the effect of a lapse mechanism downstream of spiking circuit models (*Figure 8—figure supplements 8–9*). Using the lapse rate fitted to the experimental data collected from the two monkeys, we assigned such portions of trials to have randomly selected choices for each circuit model, and repeated the analysis to obtain psychometric functions and various regression weights. Crucially, while the psychometric function as well as evidence mean and standard deviation regression weights were suppressed, the findings on PVB index were not qualitatively altered in the circuit models, further supporting the finding that the lapse rate does not account for changes in PVB under ketamine.

## Discussion

Previous studies have shown human participants exhibit choice biases when options differ in the standard deviation of the evidence samples, preferring choice options drawn from a more variable distribution (*Tsetsos et al., 2016*; *Tsetsos et al., 2012*). By utilising a behavioural task with precise experimenter control over the distributions of time-varying evidence, we show that macaque monkeys exhibit a similar pro-variance bias in their choices akin to human participants. This pro-variance bias was also present in a spiking circuit model, which demonstrated a neural mechanism for this behaviour. We then introduced perturbations at distinct synaptic sites of the circuit, which revealed dissociable predictions for the effects of NMDA-R antagonism. Ketamine produced decision-making deficits consistent with a lowering of the cortical excitation-inhibition balance.

Biophysically grounded neural circuit modelling is a powerful tool to link cellular level observations to behaviour. Previous studies have shown recurrent cortical circuit models reproduce normative decision-making and working memory behaviour, and replicate the corresponding neurophysiological activity (*Wang et al., 2013*; *Murray et al., 2014*; *Wang, 2002*; *Wong and Wang, 2006*; *Murray et al., 2017*; *Wong et al., 2007*; *Wimmer et al., 2014*). However, whether they are also capable of reproducing idiosyncratic cognitive biases has not previously been explored. Here we demonstrated pro-variance and primacy biases in a spiking circuit model. The primacy bias results from the formation of attractor states before all of the evidence has been presented. This neural implementation for bounded evidence accumulation corresponds with previous algorithmic explanations (*Kiani et al., 2008*).

The results from our spiking circuit modelling also provide a parsimonious candidate mechanism for the pro-variance bias within the evidence accumulation process. Specifically, strong evidence in favour of an option pushes the network towards an attractor state more so than symmetrically weak evidence pushes it away. Previous explanations for pro-variance bias proposed computations at the level of sensory processing upstream of evidence accumulation. In particular, a 'selective integration' model proposed that information for the momentarily weaker option is discarded before it enters the evidence accumulation process (*Tsetsos et al., 2016*). Conceptually, our model was analogous to previous models in that weak evidence is weighted less relative to strong evidence. However, there are key differences between the two models. In 'selective integration' and similar models concerning sensory processes, an asymmetric filter was applied to the stimuli before the stimuli were evaluated by the decision making process, in some upstream area that can be potentially modulated based on task demands. In contrast, in our circuit model pro-variance bias arose from the non-linearity activity profile (*Figure 6D*, see Materials and methods) of model neurons. In that sense, pro-variance bias was an intrinsic phenomenon of the evidence integration process in our circuit model.

Despite the conceptual analogy between our circuit model and the 'selective integration' model in which weak stimuli were asymmetrically weighted, our circuit model cannot be directly mapped to the latter. In the 'selective integration' model, the asymmetry is realized as a discounting of the momentarily weaker stimuli by a constant factor. In our circuit model, the asymmetry arose from the non-linearity of the transfer function. However, the transfer function was not static, but dynamically evolved with the state of the model (e.g. in the mean-field model, the transfer function depended on the two decision variables, See Materials and methods). Due to this complexity, the asymmetry of the circuit model cannot be reduced to one simple expression, and was instead closely entangled with the attractor dynamics of the system.

Crucially, our circuit model generated dissociable predictions for the effects of NMDA-R hypofunction on the pro-variance bias (PVB) index that were tested by follow-up ketamine experiments. While it remains an open question where and how in the brain the selective integration process takes place, our modelling results suggest that purely sensory deficits may not capture the alterations in choice behaviour observed under ketamine, in contrast to E/I perturbations in decision-making circuits (*Figure 7H*). Multiple complementary processes may simultaneously contribute to pro-variance bias during decision making, especially in complex behaviours over longer timescales. Future work will aim to contrast between these two models with neurophysiological data recorded while monkeys are performing this task.

On the other hand, there may also be limits in the extent to which our findings can be directly compared to those from previous studies in humans (*Tsetsos et al., 2016*; *Tsetsos et al., 2012*). For example, human studies have revealed that a 'frequent winner' bias coexists with the pro-variance

bias and may arise from the same selective integration mechanism. Unlike previous studies, our subjects did not exhibit a 'frequent winner' bias. Furthermore, although both studies demonstrate a PVB, the temporal weighting of evidence in the previous human studies exhibit recency, unlike the primacy found in the present study. This may be in part due to differences in the underlying computational regimes that are used for evidence integration, or may be due to more trivial differences between the experimental paradigms – for example, different paradigms have identified primacy (*Kiani et al., 2008*), recency (*Cheadle et al., 2014*) or noiseless sensory evidence integration (*Brunton et al., 2013*). A stronger test will be to record neurophysiological data while monkeys are performing our task; this would help to distinguish between the 'selective integration' hypothesis and the cortical circuit mechanism proposed here.

The PVB index, as the ratio of standard deviation to mean evidence regression weights, serves as a conceptually useful measure to interpret changes in pro-variance bias due to ketamine perturbation in this study. Given the model does not feature any explicit processes that mediate pro-variance bias, PVB should be understood as an emergent phenomenon arising from the decision-making process. In this context, a sensory-deficit perturbation, which down-scales the incoming evidence strength without perturbing the decision-making process, should proportionally down-scale the evidence mean and standard deviation regression weights, thus maintaining the PVB index. In contrast, lowering and elevating E/I ratio distinctly alter the dynamics of the decision-making process and thus differentially perturb the PVB index. It is also important to study how changes in the PVB index are driven by changes in the mean vs. standard deviation regression coefficients, as considering PVB index alone can obscure these effects. For instance, based on the model, the increase in PVB index by lowering E/I is generally due to a stronger decrease in mean regression coefficient than standard deviation regression coefficient (*Figure 7—figure supplement 2*). However, small perturbations of lowering E/I may actually increase PVB index due to an increase in standard deviation regression coefficient and a decrease in mean regression coefficient. As a support of this model finding, while the two monkeys both demonstrate a significant decrease in mean regression weight by ketamine, one monkey seems to demonstrate a trend to decrease standard deviation regression weight and the other seems to demonstrate a trend to increase (*Figure 8—figure supplement 2*). The two monkeys, both interpreted as lowered E/I ratio using the model-based approach in this study, may therefore experience slightly different degrees of E/I reduction when administered with ketamine, as shown through concurrent changes in NMDA-R conductances in the circuit model (*Figure 7—figure supplement 2*).

In this study we did not undertake quantitative fitting of the circuit model parameters to match the empirical data. Rather we took a previously developed circuit model and only manually adjusted input strengths to be loosely in the regime of experimental behavior. There are technical and theoretical challenges in quantitatively fitting to psychophysical behavior with biophysically-based circuit models, including reduced mean-field models, which have impeded such applications in the field. Critical challenges include computational cost of simulation, a large number of parameters with unknown effective degeneracies on behavior, and treatment of noise in mean-field reductions. Future work, beyond the scope of the present study, is needed to bridge these gaps in relating circuit models to psychophysical behavior.

Instead of direct model fitting, here we studied biophysically-based spiking circuit models for two primary purposes: to examine whether a behavioral phenomenon, such as pro-variance bias, can emerge from a regime of circuit dynamics, and through what dynamical circuit mechanisms; and to characterize how the phenomenon and underlying dynamics is altered by modulation of neurobiologically grounded parameters, such as NMDA-R conductance. The circuit modelling in this study demonstrates a set of mechanisms which is sufficient to produce the phenomenon of interest. The bottom-up mechanistic approach in this study, which makes links to the physiological effects of pharmacology and makes testable predictions for neural recordings and perturbations, is complementary to top-down algorithmic modeling approaches.

Our pharmacological intervention experimentally verified the significance of NMDA-R function for decision-making. In the spiking circuit model, NMDA-Rs expressed on pyramidal cells are necessary for reverberatory excitation, without which evidence cannot be accumulated and stable working memory activity cannot be maintained. NMDA-Rs on interneurons are necessary for maintaining background inhibition and preventing the circuit from reaching an attractor state prematurely (*Murray et al., 2014*; *Wang, 2002*). By administering ketamine, an NMDA-R antagonist, specific

short-term deficits in choice behaviour were induced, which were consistent with a lowering of the cortical excitation-inhibition balance in the circuit model. This suggests the NMDA-R antagonist we administered systemically was primarily acting to inhibit neurotransmission onto pyramidal cells and weaken the recurrent connection strength across neurons. It is important to note that in addition to the main role of ketamine as a NMDA-R antagonist, it might also target other receptor sites (*Chen et al., 2009*; *Zanos et al., 2016*; *Moaddel et al., 2013*). However, of all receptors, ketamine has by far the highest affinity for the NMDA-R receptor (*Frohlich and Van Horn, 2014*). The effects of synaptic perturbations could be interpreted in terms of their net effect on E/I balance, at least to the first order (*Murray et al., 2014*; *Lam, 2017*). For instance, in the circuit model, proportional NMDA hypofunction on both E and I neurons maintains E/I balance and minimally impairs circuit computation, while the effect of disproportionate NMDA hypofunction on E and I neurons is well captured by the direction of net change in E/I ratio (*Figure 7—figure supplements 2* and *3*). Given the highest affinity of ketamine to NMDA-Rs, the effect of NMDA-R-hypofunction should dominantly determine the direction of E/I imbalance, and should not be counter-balanced by the effect of other perturbations. Finally, other receptors and brain areas are likely altered by systemic ketamine administration, which is beyond the scope of the microcircuit model in this study.

The physiological effects of NMDA-R antagonism on in vivo cortical circuits remains an unresolved question. A number of studies have proposed a net cortical disinhibition through NMDA-R hypofunction on inhibitory interneurons (*Nakazawa et al., 2012*; *Krystal et al., 2003*; *Lisman et al., 2008*; *Lewis et al., 2012*). The disinhibition hypothesis is supported by studies finding NMDA-R antagonists mediate an increase in the firing of prefrontal cortical neurons, in rodents (*Jackson et al., 2004*; *Homayoun and Moghaddam, 2007*) and monkeys (*Ma et al., 2018*; *Ma et al., 2015*; *Skoblenick and Everling, 2012*; *Skoblenick et al., 2016*). On the other hand, the effects of NMDA-R antagonists on E/I balance may vary across neuronal sub-circuits within a brain area. For instance, in a working memory task, ketamine was found to increase spiking activity of response-selective cells, but decrease activity of the task-relevant delay-tuned cells in primate prefrontal cortex (*Wang et al., 2013*). Such specificity might explain why several studies reported less conclusive effects of NMDA-R antagonists on overall prefrontal firing rates in monkeys (*Wang et al., 2013*; *Zick et al., 2018*). In vitro work has also revealed the excitatory post-synaptic potentials (EPSPs) of prefrontal pyramidal neurons are much more reliant on NMDA-R conductance than parvalbumin interneurons (*Rotaru et al., 2011*). Other investigators combining neurophysiological recordings with modelling approaches have also concluded that the action of NMDA-R antagonists is primarily upon pyramidal cells (*Wang et al., 2013*; *Moran et al., 2015*). Our present findings, integrating pharmacological manipulation of behaviour with biophysically-based spiking circuit modelling, suggest that the ketamine-induced behavioural biases are more consistent with a lowering of excitation-inhibition balance and weakening of recurrent dynamics. Future work with electrophysiological recordings during the performance of our task, under pharmacological interventions, can potentially dissociate the effect of ketamine on E/I balance specifically in cortical neurons exhibiting decision-related signals. Notably, the decision making behaviours in our circuit model arise from attractor dynamics relying on unstructured interneurons to provide lateral feedback inhibition. Recent experiments found that, in mouse parietal cortex during a decision-making task, inhibitory parvalbumin (PV) interneurons - thought to provide feedback inhibition - may be equally selective as excitatory pyramidal neurons (*Najafi et al., 2020*). Depending on the pattern and connectivity of their feedback projections to pyramidal neurons, such a circuit structure supports different forms of evidence accumulation in cortical circuits (*Lim and Goldman, 2013*). It remains to be seen how the pro-variance bias effect and the current predictions extend to circuit models with selective inhibitory interneurons.

The minutes-long timescale of the NMDA-R mediated decision-making deficit we observed was also consistent with the psychotomimetic effects of subanaesthetic doses of ketamine in healthy humans (*Krystal et al., 1994*; *Krystal et al., 2003*). As NMDA-R hypofunction is hypothesised to play a role in the pathophysiology of schizophrenia (*Kehrer et al., 2008*; *Olney and Farber, 1995*; *Krystal et al., 2003*; *Lisman et al., 2008*), our findings have important clinical relevance. Previous studies have demonstrated impaired perceptual discrimination in patients with schizophrenia performing the random-dot motion (RDM) decision-making task (*Chen et al., 2003*; *Chen et al., 2004*; *Chen et al., 2005*). Although RDM tasks have been extensively used to study evidence accumulation (*Gold and Shadlen, 2007*), previously this performance deficit in schizophrenia was interpreted as

reflecting a diminished representation of sensory evidence in visual cortex (*Chen et al., 2003*; *Butler et al., 2008*). Based on our task with precise temporal control of the stimuli, our findings suggest that NMDA-R antagonism alters the decision-making process in association cortical circuits. Dysfunction in these association circuits may therefore provide an important contribution to cognitive deficits - one that is potentially complementary to upstream sensory impairment. Crucially, our task uniquely allowed us to rigorously verify that the subjects used an accumulation strategy to guide their choices (cf. previous animal studies [*Gold and Shadlen, 2007*; *Roitman and Shadlen, 2002*; *Hanks et al., 2015*; *Morcos and Harvey, 2016*; *Katz et al., 2016*]), with these analyses suggesting the strategy our subjects employed was relatively consistent with findings in human participants. This consistency further ensures our findings may translate across species, in particular to clinical populations.

Another related line of schizophrenia research has shown a decision-making bias known as jumping to conclusions (JTC) (*Ross et al., 2015*; *Huq et al., 1988*). The JTC has predominately been demonstrated in the 'beads task', a paradigm where participants are shown two jars of beads, one mostly pink and the other mostly green (typically 85%). The jars are hidden, and the participants are presented a sequence of beads drawn from a single jar. Following each draw, they are asked if they are ready to commit to a decision about which jar the beads are being drawn from. Patients with schizophrenia typically make decisions based on fewer beads than controls. Importantly, this JTC bias has been proposed as a mechanism for delusion formation. Based on the JTC literature, one plausible hypothesis for behavioural alteration under NMDA-R antagonism in our task may be a strong increase in the primacy bias, whereby only the initially presented bar samples would be used to guide the subjects' decisions. However, following ketamine administration, we did not observe a strong primacy – instead all samples received roughly the same weighting. There are important differences between our task and the beads task. In our task, the stimulus presentation is shorter (2 s, compared to slower sampling across bead draws), and is of fixed duration rather than terminated by the subject's choice, and therefore may not involve the perceived sampling cost of the beads task (*Ermakova et al., 2019*).

Our precise experimental paradigm and complementary modelling approach allowed us to meticulously quantify how monkeys weight time-varying evidence and robustly dissociate sensory and decision-making deficits – unlike prior studies using the RDM and beads tasks. Our approach can be readily applied to experimental and clinical studies to yield insights into the nature of cognitive deficits and their potential underlying E/I alterations in pharmacological manipulations and pathophysiologies across neuropsychiatric disorders, such as schizophrenia (*Wang and Krystal, 2014*; *Huys et al., 2016*) and autism (*Wang and Krystal, 2014*; *Yizhar et al., 2011*; *Lee et al., 2017*; *Marín, 2012*). Finally, our study highlights how precise task design, combined with computational modelling, can yield translational insights across species, including through pharmacological perturbations, and across levels of analysis, from synapses to cognition.

## Materials and methods

### Subjects

Two adult male rhesus monkeys (*M. mulatta*), subjects A and H, were used. The subjects weighed 12–13.3 kg, and both were ~6 years old at the start of the data collection period. We regulated their daily fluid intake to maintain motivation in the task. All experimental procedures were approved by the UCL Local Ethical Procedures Committee and the UK Home Office (PPL Number 70/8842), and carried out in accordance with the UK Animals (Scientific Procedures) Act.

### Behavioural protocol

Subjects sat head restrained in a primate behavioural chair facing a 19-inch computer screen (1,280 × 1024 px screen resolution, and 60 Hz refresh rate) in a dark room. The monitor was positioned 59.5 cm away from their eyes, with the height set so that the centre of the screen aligned with neutral eye level for the subject. Eye position was tracked using an infrared camera (ISCAN ETL-200) sampled at 240 Hz. The behavioural paradigm was run in the MATLAB-based toolbox MonkeyLogic (http://www.monkeylogic.net/, Brown University) (*Asaad and Eskandar, 2008a*; *Asaad and Eskandar, 2008b*; *Asaad et al., 2013*). Eye position data were relayed to MonkeyLogic for use

online during the task, and was recorded for subsequent offline analysis. Following successful trials, juice reward was delivered to the subject using a precision peristaltic pump (ISMATEC IPC). Subjects performed two types of behavioural sessions: standard and pharmacological. In pharmacological sessions, following a baseline period, either an NMDA-R antagonist (Ketamine) or saline was administered via intramuscular injection. Monkey A completed 41 standard sessions, and 28 pharmacological sessions (15 ketamine; 13 saline). Monkey H completed 68 standard sessions, and 35 pharmacological sessions (18 ketamine; 17 saline).

## Injection protocol

Typically, two pharmacological sessions were performed each week, at least 3 days apart. Subjects received either a saline or ketamine injection into the trapezius muscle while seated in the primate chair. Approximately 12 min into the session, local anaesthetic cream was applied to the muscle. At 28 min, the injection was administered. The task was briefly paused for this intervention (64.82 ± 10.85 secs). Drug dose was determined through extensive piloting, and a review of the relevant literature (*Wang et al., 2013*; *Blackman et al., 2013*). The dose used was 0.5 mg/kg.

## Task

Subjects were trained to perform a two-alternative value-based decision-making task. A series of bars, each with different heights, were presented on the left and right-side of the computer monitor. Following a post-stimulus delay, subjects were rewarded for saccading towards the side with either the taller or shorter average bar-height, depending upon a contextual cue displayed at the start of the trial (see *Figure 1A* inset). The number of pairs of bars in each series was either four ('*4Sample-Trial*') or eight ('*8SampleTrial*') during trials in each standard behavioural session. In this report, we only consider the results from the eight sample trials, though similar results were obtained from the four sample trials. The number of bars was always six during pharmacological sessions.

The bars were presented inside of fixed-height rectangular placeholders (width, 84px; height, 318px). The placeholders had a black border (thickness 9px), and a grey centre where the stimuli were presented (width, 66px; height, 300px). The bar heights could take discrete percentiles, occupying between 1% and 99% of the grey space. The height of the bar was indicated by a horizontal black line (thickness 6px). Beneath the black line, there was 45° grey gabor shading.

An overview of the trial timings is outlined in *Figure 1A*. Subjects initiated a trial by maintaining their gaze on a central, red fixation point for 750 ms. After this fixation was completed, one of four contextual cues (see *Figure 1A* inset) was centrally presented for 350 ms. Subjects had previously learned that two of these cues instructed to choose the side with the taller average bar-height ('*ChooseTall*' trial), and the other two instructed to choose the side with the shorter average bar-height ('*ChooseShort*' trial). Next, two black masks (width, 84px; height, 318px) were presented for 200 ms in the location of the forthcoming bar stimuli. These were positioned either side of the fixation spot (6° visual angle from centre). Each bar stimulus was presented for 200 ms, followed by a 50 ms inter-stimulus-interval where only the fixation point remained on the screen. Once all of the bar stimuli had been presented, the mask stimuli returned for a further 200 ms. There was then a post stimulus delay (250–750 ms, uniformly sampled across trials). Following this, the colour of the fixation point was changed to green (go cue), and two circular saccade targets appeared on each side of the screen where the bars had previously been presented. This cued the subject to indicate their choice by making a saccade to one of the targets. Once the subject reported their decision, there were two stages of feedback. Immediately following choice, the green go cue was extinguished, the contextual cue was re-presented centrally, along with the average bar heights of the two series of stimuli previously presented. The option the subject chose was indicated by a purple outline surrounding the relevant bar placeholder (width, 3.8°; height, 10°). Following 500 ms, the second stage of feedback began. The correct answer was indicated by a white outline surrounding the bar placeholder (width, 5.7°; height, 15°). On correct trials, the subject was rewarded for a length of time proportional to the average height of the chosen option (directly proportional on a '*ChooseTall*' trial, negatively proportional on a '*ChooseShort*' trial). On incorrect trials, there was no reward. Regardless of the reward amount, the second feedback stage lasted 1200 ms. This was followed by an inter-trial-interval (1.946 ± 0.051 secs; for Standard Sessions, across all completed included trials). The inter-trial-

interval duration was longer on *'4SampleTrials'* than '8SampleTrials', in order for the trials to be an equal duration, and facilitate a similar reward rate between the two conditions.

Subjects were required to maintain central fixation from the fixation period until they indicated their choice. If the initial fixation period was not completed, or fixation was subsequently broken, the trial was aborted and the subject received a 3000 ms timeout (Trials in standard sessions: Monkey A – 22.46%, Monkey H – 15.27%). On the following trial, the experimental condition was not repeated. If subjects failed to indicate their choice within 8000 ms, a 5000 ms timeout was initiated (Trials in standard sessions: Monkey A - 0%, Monkey H – 0%).

Experimental conditions were blocked according to the contextual cue and evidence length. This produced four block types (*ChooseTall4SampleTrial (T4), ChooseTall8SampleTrial (T8), Choose-Short4SampleTrial (S4), ChooseShort8SampleTrial (S8)*). At the start of each session, subjects performed a short block of memory-guided saccades (MGS) (*Hikosaka and Wurtz, 1983*), completing 10 trials. Data from these trials are not presented in this report. Following the MGS block, the first block of decision-making trials was selected at random. After the subject completed 15 trials in a block, a new block was selected without replacement. Each new block had to have either the same evidence length or the same contextual cue as the previous block. After all four blocks had been completed, there was another interval of MGS trials. A new evidence accumulation start block was then randomly selected. As there were four block types, and either the evidence length or the contextual cue had to be preserved across a block switch, there were two 'sequences' in which the blocks could transition (i.e. T4→T8→S8→S4; or T4→S4→S8→T8, if starting from T4). Following the intervening MGS trials, the blocks transitioned in the opposite sequence to those used previously, starting from the new randomly chosen block. This block switching protocol was continued throughout the session. At the start of each block, the background of the screen was changed for 5000 ms to indicate the evidence length of the forthcoming block. A burgundy colour indicated an eight sample block was beginning, a teal colour indicated a four sample block was beginning.

## Trial generation

The heights of the bars on each trial were precisely controlled. On the majority of trials (Regular Trials, Completed trials in standard sessions: Monkey A – 76.67%, Monkey H – 76.23%), the heights of each option were generated from independent Gaussian distributions (*Figure 4A-B*). There were two levels of variance for the distributions, designated as 'Narrow' and 'Broad'. The mean of each distribution, μ, was calculated as $\mu = 50 + Z \times \sigma$, where $Z \sim \mathcal{U}(-0.25, 0.25)$, and $\sigma$ was either 12 or 24 for narrow and broad stimuli streams. The individual bar heights were then determined by $\sim \mathcal{N}(\mu, \sigma)$. The trial generation process was constrained so the samples reasonably reflected the generative parameters. These restrictions required bar heights to range from 1 to 99, and the actual σ for each stream to be no more than 4 from the generative value. On any given trial, subjects could be presented with two narrow streams, two broad streams, or one of each. The evidence variability was therefore independent between the two streams. For post-hoc analysis (*Figure 4*) we defined one stream as the 'Lower SD' option on each trial, and the other the 'Higher SD' option, based upon the sampled/actual $\sigma$.

A proportion of 'decision-bias trials' were also specifically designed to elucidate the effects of evidence variability on choice, and whether subjects displayed primacy/recency biases (*Tsetsos et al., 2012*). These trials occurred in equal proportions within all four block types. Only one of these decision-bias trial types was tested in each behavioural session.

Narrow-broad trials (Completed trials in standard sessions: Monkey A – 14.87%, Monkey H – 15.78%) probed the effect of evidence variability on choice (*Tsetsos et al., 2012*). Within this category of trials, there were three conditions (*Figure 3A*). In each, the bar heights of one alternative were associated with a narrow Gaussian distribution ($\sim \mathcal{N}(\mu_N, 12)$), and the bar heights from the other with a broad Gaussian distribution ($\sim \mathcal{N}(\mu_B, 24)$). In the first two conditions, *'Narrow Correct'* ($\mu_N \sim \mathcal{U}(48, 60)$, $\mu_B = \mu_N - 8$) and *'Broad Correct'* ($\mu_B \sim \mathcal{U}(48, 60)$, $\mu_N = \mu_B - 8$), there was a clear correct answer. In the third condition, *'Ambiguous'* ($\mu_B \sim \mathcal{U}(44, 56)$, $\mu_N = \mu_B$), there was only small evidence in favour of the correct answer. In all of these conditions, the generated samples had to be within 4 of the generating σ. Furthermore, on 'Narrow Correct' and 'Broad Correct' trials the difference between the mean evidence of the intended correct and incorrect stream had to range from +2 to +14. On the 'Ambiguous' trials, the mean evidence in favour of one option over the other was constrained to be <4. A visualisation of the net evidence in each of these trial types is displayed

(*Figure 3A*). For the purposes of illustration, the probability density was smoothed by a sliding window of ±1, within the generating constraints described above ('Narrow Correct' and 'Broad Correct' trials have net evidence for correct option within [2, 14]; 'Ambiguous' trials have net evidence within [-4, 4]). A very small number of trials were excluded from this visualisation, because their net evidence fell marginally outside the constraints. This was because bar heights were rounded to the nearest integer (due to the limited number of pixels on the computer monitor) after the generating procedure and the plot reflects the presented bar heights.

Half-half trials (Completed trials in standard sessions: Monkey A – 8.46%, Monkey H – 8.00%) probed the effect of temporal weighting biases on choice (*Tsetsos et al., 2012*). The heights of each option were generated using the same Gaussian distribution (X $\sim \mathcal{N}$ ($\mu_{HH}$, 12), where $\mu_{HH} \sim \mathcal{U}$ (40, 60)). This distribution was truncated to form two distributions: $X_{Tall}$ {mean(X) – 0.5*SD(X),∞}, and $X_{Short}$ {-∞, mean(X) + 0.5*SD(X)}. On each trial, one option was designated '*TallFirst*' – where the first half of bar heights was drawn from $X_{Tall}$ and the second half of bar heights drawn from $X_{Short}$. This process was also constrained so that the mean of samples drawn from $X_{Tall}$ had to be at least 7.5 greater than those taken from $X_{Short}$. The other option was '*ShortFirst*', where the samples were drawn from the two distributions in the reverse order.

## Task modifications for pharmacological sessions

Minor adjustments were made to the task during the pharmacological sessions to maximise trial counts available for statistical analysis. Trial length was fixed to 6 pairs of samples. The block was switched between '*ChooseTall6Sample*' and '*ChooseShort6Sample*' after 30 completed trials, without intervening MGS trials. From our pilot data, it was clear ketamine reduced choice accuracy. In order to maintain subject motivation, the most difficult 'Regular' and 'HalfHalf' trials were not presented. Following the trial generation procedures described above, in pharmacological sessions these trials were additionally required to have >4 mean difference in evidence strength. Of the '*Narrow-Broad*' trials, only '*Ambiguous*' conditions were used; but no further constraints were applied to these trials. In some sessions, a small number of control trials were used, in which the bar heights for each option were fixed across all of the samples. All analyses utilised 'Regular', 'Half-Half', and 'Narrow-Broad' trials. Monkey H did not always complete sufficient trials once ketamine was administered. Sessions where the number of completed trials was fewer than the minimum recorded in the saline sessions were discarded (6 of 18 sessions). Following ketamine administration, Monkey A did not complete fewer trials in any session than the minimum recorded in a saline session.

## Behavioural data analysis

To assess decision-making accuracy during standard sessions, we initially fitted a psychometric function (*Kiani et al., 2008*; *Roitman and Shadlen, 2002*) to subjects' choices pooled across 'Regular' and 'Narrow-Broad' trials (*Figure 2A-B*). This defines the choice accuracy (*P*) as a function of the difference in mean evidence in favour of the correct choice (evidence strength,*x*):

$$P(x) = 0.5 + 0.5 \left( 1 - exp \left( - \left( \frac{x}{\alpha} \right)^{\beta} \right) \right) \qquad (1)$$

where $\alpha$ and $\beta$ are respectively the discrimination threshold and order of the psychometric function, and $exp$ is the exponential function. To illustrate the effect of pro-variance bias, we also fitted a three-parameter psychometric function to the subjects' probability to choose the higher SD option ($P_{HSD}$) in the 'Regular' trials, as a function of the difference in mean evidence in favour of the higher SD option on each trial ($x_{HSD}$):

$$P_{HSD}(x_{HSD}) = 0.5 + 0.5 \, sign \, (x_{HSD} + \delta) \left( 1 - exp \left( - \left( \frac{|x_{HSD} + \delta|}{\alpha} \right)^{\beta} \right) \right) \qquad (2)$$

where $\delta$ is the psychometric function shift, and *sign* returns 1 and -1 for positive and negative inputs respectively. To be explicitly clear, on '*ChooseTall*' trials, the mean evidence in favour of the higher SD option was calculated by subtracting the mean bar height of the lower SD option from that of the higher SD option. On '*ChooseShort*' trials, the mean evidence in favour of the higher SD option

was calculated by subtracting [100 - mean bar height of the lower SD option] from [100 – mean bar height of the higher SD option].

In both cases, the psychometric function is fitted using the method of maximum-likelihood estimation (MLE), with the estimator

$$\sum_i [\mathbb{1}_i * log(P(\mathrm{x})) + (1 - \mathbb{1}_i) * log(1 - P(\mathrm{x}))] \tag{3}$$

(and similarly for $P_{HSD}$ & $x_{HSD}$), where $i$ is summed across trials. $\mathbb{1}_i = 1$ if the correct (higher SD) option is chosen in trial $i$ and 0 otherwise.

The temporal weights of stimuli were calculated using logistic regression. This function defined the probability ($P_L$) of choosing the left option:

$$ln\left(\frac{P_L}{1 - P_L}\right) = \beta_0' + \sum_{n=1}^{8} \beta_n'(L_n - R_n) \tag{4}$$

where $\beta_0'$ is a bias term, $\beta_n'$ reflects the weighting given to the nth pair of stimuli, $L_n$ and $R_n$ reflect the evidence for the left and right option at each time point.

Regression analysis was used to probe the influence of evidence mean, and evidence variability on choice during the 'Regular' trials (**Figures 4D, 5F, 6C, 7F–H** and **8D-F**, **Figure 4—figure supplement 1D,G**, **Figure 8—figure supplement 1C,H**). This function defined the probability ($P_L$) of choosing the left option:

$$ln\left(\frac{P_L}{1 - P_L}\right) = \beta_0 + \beta_1(mean(L) - mean(R)) + \beta_2\,(std(L) - std(R)) \tag{5}$$

where $\beta_0$ is a bias term, $\beta_1$ reflects the influence of evidence mean, and $\beta_2$ reflects the influence of standard deviation of evidence (evidence variability).

This approach was extended to probe other potential influences on the decision-making process. An expanded regression model was defined as follows:

$$\begin{aligned}
ln\left(\tfrac{P_L}{1 - P_L}\right) &= \beta_0 + \beta_1(mean(L)) + \beta_2\,(std(L)) + \beta_3\,(Max(L)) \\
&+ \beta_4\,(Min(L)) + \beta_5\,(L_1) + \beta_6\,(L_8) + \beta_7\,(mean(R)) + \beta_8\,(std(R)) \\
&+ \beta_9\,(Max(R)) + \beta_{10}\,(Min(R)) + \beta_{11}\,(R_1) + \beta_{12}\,(R_8)
\end{aligned} \tag{6}$$

where $\beta_0$ is a bias term, $\beta_1$ reflects the influence of evidence mean of the left samples, $\beta_2$ reflects the influence of evidence variability of the left samples, $\beta_3$ reflects the influence of the maximum left sample, $\beta_4$ reflects the influence of the minimum left sample, $\beta_5$ reflects the influence of the first left sample, $\beta_6$ reflects the influence of the last left sample. $\beta_7$ to $\beta_{12}$ reflect the same attributes for samples on the right side of the screen. Due to strong correlations among evidence standard deviation, maximum, and minimum, the regression model without $\beta_2$ and $\beta_8$ is used to evaluate the contribution of regressors other than evidence mean and standard deviation to the decision making process (**Figure 4—figure supplement 1E,H**, **Figure 5—figure supplement 1B**, **Figure 6—figure supplement 1B**, **Figure 7—figure supplement 1B**, **Figure 8—figure supplement 1D,I**).

To explore whether the subjects demonstrated a frequent-winner bias (**Tsetsos et al., 2012**), whereby they prefer to choose options that more frequently have the greater evidence across samples, we used a regression approach (**Figure 4—figure supplement 3**). The regression equation defined the probability (PL) of choosing the left option:

$$\ln\left(\frac{P_L}{1 - P_L}\right) = \beta_0 + \beta_1\,(mean(L) - mean(R)) + \beta_2\,(LocalWins(L) - LocalWins(R)) \tag{7}$$

where $\beta_0$ is a bias term, $\beta_1$ reflects the influence of evidence mean, and $\beta_2$ reflects the influence of local winners (frequent-winner bias). The number of local wins for each option ranges between 0 and 8, and is the amount of times that the momentary evidence is stronger for that option. To provide an example, consider a trial where the evidence values were Left: [50 55 56 48 80 45 30 50], Right: [55 48 90 34 70 50 50 70]. Here, there would be 3 local wins for the left option, and 5 local wins for the right option.

To control for possible lapse effects induced by ketamine, where the animal responded randomly regardless of the trial difficulty, the behavioural models described above were extended to include an extra 'lapse parameter', $Y_0$. The purpose of this parameter was to quantify the frequency of lapses, and to isolate the effect of lapsing from our other analyses of interest (i.e. the effect of ketamine on PVB index). In other words, lapse rate refers to the asymptote error rate at the limit of strong evidence. *Equations 4-6* were extended as follows:

$$ln\left(\frac{P_L - Y_0}{1 - P_L - Y_0}\right) = \beta_0' + \sum_{n=1}^{8}\beta_n'(L_n - R_n) \tag{8}$$

$$\ln\left(\frac{P_L - Y_0}{1 - P_L - Y_0}\right) = \beta_0 + \beta_1(mean(L) - mean(R)) + \beta_2\,(std(L) - std(R)) \tag{9}$$

$$
\begin{aligned}
ln\left(\frac{P_L - Y_0}{1 - P_L - Y_0}\right) &= \beta_0 + \beta_1(mean(L)) + \beta_2\,(std(L)) + \beta_3\,(Max(L)) \\
&+ \beta_4\,(Min(L)) + \beta_5\,(L_1) + \beta_6\,(L_8) + \beta_7\,(mean(R)) \\
&+ \beta_8\,(std(R)) + \beta_9\,(Max(R)) + \beta_{10}\,(Min(R)) + \beta_{11}\,(R_1) + \beta_{12}\,(R_8)
\end{aligned}
\tag{10}
$$

The models including a lapse term (*Equations 8-10*) were fitted via maximum-likelihood estimation (using the *fminsearch* algorithm in MATLAB), using the following cost function:

$$\Sigma_i[\mathbb{1}_i * log(P(\mathrm{x}_i)) + (1 - \mathbb{1}_i) * log(1 - P(\mathrm{x}_i))] + \lambda * \left(\sum_{j=0}^{M}\beta_j^2 + Y_0^2\right) \tag{11}$$

where $i$ is summed across trials. $\mathbb{1}_i = 1$ if the left option is chosen in trial $i$ and 0 otherwise. $\lambda$ is an L2 regularisation constant, which was set to 0.01. Bootstrapping was used to generate error estimates for the parameters of these models (*10,000 iterations*). As our analyses demonstrate that the animals very rarely lapse when administered with saline, we did not deem it necessary to apply the lapsing models to the standard session experiment (i.e. *Figures 2, 3, 4, 5, 6*).

To visualise the influence of lapsing upon the psychometric functions, and to allow a comparison between the monkey behaviour and circuit model performance, we extended *Equation 2*:

$$P_{HSD}(x_{HSD}) = 0.5 + 0.5(1 - 2Y_0)\,sign\,(x_{HSD} + \delta)\left(1 - exp\left(-\left(\frac{|x_{HSD} + \delta|}{\alpha}\right)^{\beta}\right)\right) \tag{12}$$

Here, $Y_0$ was a fixed parameter according to the lapse rate calculated from the relevant monkey's behavioural data.

The goodness-of-fit of various regression models with combinations of the predictors in the full model (*Equation 6*) were compared using a 10-fold cross-validation procedure (*Supplementary files 1–4*). Trials were initially divided into 10 groups. Data from 9 of the groups were used to train each regression model and calculate regression coefficients. The likelihood of the subjects' choices in the left-out group (testing group), given the regression coefficients, could then be determined. The log-likelihood was then summed across these left-out trials. This process was repeated so that each of the 10 groups acted as the testing group. The whole cross-validation procedure was performed 100 times, and the average log-likelihood values were taken.

To initially explore the time course of drug effects on decision-making, we plotted choice accuracy (combined across 'Regular', 'Half-Half' and 'Narrow-Broad' trials) relative to drug administration (*Figure 8A*). Trials were binned relative to the time of injection. Within each session, choice accuracy was estimated at every minute, using a 6 min window around the bin centre. Accuracy was then averaged across sessions. To further probe the influence of drug administration on decision-making, we defined an analysis window based upon the time course of behavioural effects. All trials before the time of injection were classified as 'pre-drug'. All trials beginning 5–30 min after injection were defined as 'on-drug' trials. These trials were then analysed using the same methods as described for the Standard sessions.

To quantify the effect of ketamine administration on the PVB index (*Figure 8F*, *Figure 8—figure supplement 1C,H*), we performed a permutation test. Trials collected during ketamine

administration were compared with those collected during saline administration. The test statistic was calculated as the difference between the PVB index in ketamine and saline conditions. For each permutation, trials from the two sets of data were pooled together, before two shuffled sets with the same number of trials as the original ketamine and saline data were extracted. Next, the PVB index was computed in each permuted set, and the difference between the two PVB indices calculated. The difference measure for each permutation was used to build a null distribution with 1000000 entries. The difference measure from the true data were compared with the null distribution to calculate a p-value. For the models including a lapse term (*Figure 8—figure supplement 2*), the same test was performed with 10,000 permutations.

We later revisited the time course of drug effects by running our regression analyses at each of the binned windows described above (*Figure 8—figure supplement 3*). To calculate the time window where a parameter differed between ketamine and saline conditions, we used a cluster-based permutation test (*Nichols and Holmes, 2002*; *Cavanagh et al., 2018*; *Cavanagh et al., 2016*). These tests allowed us to correct for multiple comparisons while assessing the significance of time series data. The difference between the parameter of interest (PVB index) was calculated in the true data for each timepoint. All consecutive timepoints when this statistic exceeded a threshold $((|\mathrm{PVB}_{\mathrm{Saline}} - \mathrm{PVB}_{\mathrm{ketamine}}| \geq 0.15))$ were designated as a 'cluster'. The size of the clusters were compared to a null distribution constructed using a permutation test. The drug administered (ketamine or saline) in each session was randomly permuted 10,000 times and the cluster analysis was repeated for each permutation. The size of the largest cluster for each permutation was entered into the null distribution. The true cluster size was significant at the p < 0.05 level if the true cluster length exceeded the 95th percentile of the null distribution.

## Spiking circuit model

A biophysically-based spiking circuit model was used to replicate decision making dynamics in a local association cortical microcircuit. The model was based on *Wang, 2002*, but with minor modifications from a previous study (*Lam, 2017*). The current model had one extra change in the input representation of the stimulus, described in detail below.

The circuit model consisted of $N_E = 1600$ excitatory pyramidal neurons and $N_I = 400$ inhibitory interneurons, all simulated as leaky integrate-and-fire neurons. All neurons were recurrently connected to each other, with NMDA and AMPA conductances mediating excitatory connections, and GABA$_A$ conductances mediating inhibitory connections. All neurons also received background inputs, while selective groups of excitatory neurons (see below) received stimulus inputs. Both background and stimulus inputs were mediated by AMPA conductances with Poisson spike trains.

Within the population of excitatory neurons were two non-overlapping groups of size $N_{E,G} = 240$. Neurons within the two groups received separate inputs reflecting the left and right stimuli streams. Neurons in the same group preferentially connected to each other (with a multiplicative factor $w_+>1$ to the connection strength), allowing integration of the stimulus input. The connection strength to any other excitatory neurons was reduced by a factor $w_-<1$ in a manner which preserved the total connection strength. Due to lateral inhibition mediated by interneurons, excitatory neurons in the two different groups competed with each other. Inhibitory neurons, as well as excitatory neurons not in the two groups, were insensitive to the presented stimuli and were non-selective toward either choices or the respective neuron groups.

Momentary stimuli bar evidences were modelled as Poisson inputs (from an upstream sensory area) to the two groups of excitatory neurons (*Figure 5A*). The mean rate of Poisson input for any group, μ, linearly scaled with the corresponding stimulus evidence:

$$\mu = \mu_0 + \mu'(h - 50) \tag{13}$$

where $h \in [0, 100]$ represented the momentary stimulus evidence, equal to the bar height in '*Choose-Tall*' trials, and 100 minus bar height in '*ChooseShort*' trials. $\mu_0 = 30Hz$ was the input strength when $h = 50$, and $\mu' = 1Hz$. For simplicity, we assumed each bar stimulus lasted 250ms, rather than 200ms with a subsequent 50ms inter-stimuli interval as in the experiment.

The circuit model simulation outputs spike data for the two excitatory populations, which are then converted to population activity smoothened with a 0.001s time-step via a casual exponential filter. In particular, for each spike of a given neuron, the histogram-bins corresponding to times

before that spike receives no weight, while the histogram-bins corresponding to times after the spike receives a weight of $\frac{1}{\tau_{\text{filter}}}\exp\left(\frac{-\Delta t}{\tau_{\text{filter}}}\right)$, where $\Delta t$ is the time of the histogram-bin after the spike, and $\tau_{\text{filter}} = 20\text{ms}$.

From the population activity of the two excitatory populations, a choice is selected 2 s after stimulus offset, based on the population with higher activity. Stimulus inputs in general drive categorical, winner-take-all competitions such that the winning population will ramp up its activity until a high attractor state (>30 Hz, in comparison to approximately 1.5 Hz baseline firing rate), while suppressing the activity of the other population below baseline via lateral inhibition (*Figure 5B*). It is also possible that neither population reaches the high-activity state. Both populations, remaining at the spontaneous state, will have similarly low activities, such that the decision readout is random.

In addition to the control model, three perturbed spiking circuit models were considered (*Murray et al., 2014*; *Lam, 2017*): lowered E/I balance, elevated E/I balance, and sensory deficit. E/I perturbations were implemented through hypofunction of NDMARs (*Figure 7A*), as this is a leading hypothesis in the pathophysiology of schizophrenia (*Nakazawa et al., 2012*; *Kehrer et al., 2008*; *Lisman et al., 2008*). NMDA-R antagonists such as ketamine also provide a leading pharmacological model of schizophrenia (*Krystal et al., 1994*; *Krystal et al., 2003*). NMDA-R hypofunction on excitatory neurons (reduced $G_{E \to E}$) resulted in lowered E/I ratio, whereas NMDA-R hypofunction on interneurons (reduced $G_{E \to I}$) resulted in elevated E/I ratio due to disinhibition (*Lam, 2017*). Sensory deficit was implemented as weakened scaling of external inputs to stimuli evidence, resulting in a reduced $\mu'$. For the exact parameters, the lowered E/I model reduced $G_{E \to E}$ by 1.3125%, the elevated E/I model reduced $G_{E \to I}$ by 2.625%, and the sensory deficit model had a sensory deficit of 20% (such that $\mu'$ was reduced by 20%) (*Figure 7*, *Figure 7—figure supplement 1*). The $G_{E \to E}$ reduction parameter was chosen as the perturbation strength which fits most well to the effect of ketamine on monkey behavioural alteration (*Figure 8—figure supplements 4*, *5*). The $G_{E \to I}$ reduction and sensory deficit parameters were chosen to match the reduction of mean evidence regression coefficient in the $G_{E \to E}$ perturbation (*Figure 7—figure supplements 2*, *4*).

The control circuit model completed 94,000 'Regular' trials, where both streams were narrow in 25% of the trials, both streams were broad in 25% of the trials, and one stream was narrow and one was broad in 50% of the trials (*Figure 5*, *Figure 5—figure supplements 1* and *2*). All trials were generated identically as in standard session experiments. The control model also completed 47,000 standard session Narrow-Broad trials. To evaluate the effect of circuit perturbations, the control model, the lowered E/I model, the elevated E/I model, and the sensory deficit model all completed an identical set of 40,000 'Regular' trials, where both streams were narrow in 25% of the trials, both streams were broad in 25% of the trials, and one stream was narrow and one was broad in 50% of the trials (*Figure 7*, *Figure 7—figure supplement 1*). The same permutation test described earlier for comparing PVB index between ketamine and saline conditions was also used to quantify whether various perturbed circuit models have different PVB indices relative to the control model (*Figure 7H*).

## Testing the versatility of model predictions

To examine the versatility of the model predictions by perturbations on the pro-variance bias effect, we parametrically reduced both $G_{E \to E}$ and $G_{E \to I}$ concurrently, by {0%, 0.4375%, 0.875%, 1.3125%, 1.75%, 2.1875%, 2.625%} for $G_{E \to E}$ and {0%, 0.875%, 1.75%, 2.625%, 3.5%, 4.375%, 5.25%} for $G_{E \to I}$ (*Figure 7—figure supplements 2*, *3*). In addition, we also parametrically varied the sensory deficit, with a sensory deficit of {0%, 5%, 10%, 15%, 20%, 25%, 30%, 35%, 40%, 45%, 50%} (*Figure 7—figure supplement 4*). 12,000 'Regular trials' were completed for each condition in the parameter scans, with the same distribution of narrow/broad streams as in the four main circuit models.

The effect of various perturbations to the circuit model was compared to the ketamine effect on the choice behaviour of the two monkeys, using coefficients from the regression model with left-right difference in mean evidence and evidence standard deviation as regressors (*Equation 5*). In particular, for each perturbation condition, the relative difference in mean evidence regression coefficient between the perturbed circuit model and the control model, and the relative difference in evidence standard deviation regression coefficient between the perturbed circuit model and the control model, were computed. Similarly, the relative differences in the two regression coefficients between the monkey data under ketamine vs saline injection were also computed (with lapse rate

accounted for). The direction of alterations was mapped to the 2-dimensional space of relative coefficient differences for mean evidence and evidence standard deviations, and was compared between the perturbations to model and monkey choice behaviour using cosine similarity (*CS*) and Euclidean distance (*ED*) (*Figure 8—figure supplements 4* and *5*):

$$CS = \frac{\delta\beta_{mean}^{monkey}\, \delta\beta_{mean}^{model} + \delta\beta_{std}^{monkey}\, \delta\beta_{std}^{model}}{\sqrt{\delta\beta_{mean}^{monkey\,2} + \delta\beta_{std}^{monkey\,2}}\, \sqrt{\delta\beta_{mean}^{model\,2} + \delta\beta_{std}^{model\,2}}} \tag{14}$$

$$ED = \sqrt{\left(\delta\beta_{mean}^{monkey} - \delta\beta_{mean}^{model}\right)^2 + \left(\delta\beta_{std}^{monkey} - \delta\beta_{std}^{model}\right)^2} \tag{15}$$

$$\delta\beta_{mean}^{monkey} = \frac{\left(\beta_{mean}^{monkey\,ketamine} - \beta_{mean}^{monkey\,saline}\right)}{\beta_{mean}^{monkey\,saline}}, \delta\beta_{mean}^{model} = \frac{\left(\beta_{mean}^{model\,pert.} - \beta_{mean}^{model\,control}\right)}{\beta_{mean}^{model\,control}}$$
$$\delta\beta_{std}^{monkey} = \frac{\left(\beta_{std}^{monkey\,ketamine} - \beta_{std}^{monkey\,saline}\right)}{\beta_{std}^{monkey\,saline}}, \delta\beta_{std}^{model} = \frac{\left(\beta_{std}^{model\,pert.} - \beta_{std}^{model\,control}\right)}{\beta_{std}^{model\,control}}$$

where the subscript denoted the regression coefficient (mean evidence or evidence standard deviation), the superscript denoted the data of the regression analysis (monkey under ketamine injection, monkey under saline injection, control circuit model, or the model with the perturbation condition of interest). A higher cosine similarity (and lower Euclidean distance) meant the relative extent (and direction) of alteration, to the regression coefficients of mean evidence and evidence standard deviation, was more similar between the perturbations in the circuit model and the monkey data.

In contrast to the two measures above which evaluate the effect of perturbation (e.g. by ketamine), Kullback–Leibler (KL) divergence allows direct comparison between monkey behavior under saline or ketamine injections, and various model conditions. More explicitly, for each monkey's data collected under the influence of ketamine or saline, and for each model condition in the parameter space, we computed the KL divergence of the choice behaviors from the model to the monkey data.

$$D_{KL} = \sum_{x_{HSD}} P_{monkey}(x_{HSD}) log\left(\frac{P_{monkey}(x_{HSD})}{P_{model}(x_{HSD})}\right) \tag{16}$$

where $x_{HSD}$ is summed over the range of net evidence strength in favour of the higher SD option on each trial, while $P_{monkey}$ and $P_{model}$ are the choice probabilities for the monkey and model to respectively choose the broad option given $x_{HSD}$.

## Mean field model

The current spiking circuit model was mathematically reduced to a mean-field model, as outlined in *Niyogi and Wong-Lin, 2013*, in the same manner as from *Wang, 2002* to *Wong and Wang, 2006*. The mean-field model consisted of two variables ($S_1$, $S_2$), namely the NMDA-R gating variables of the two groups of excitatory neurons representing the integrated evidence for the two choices. The two gating variables evolved according to:

$$\frac{dS_i}{dt} = -\frac{S_i}{\tau_{NMDA}} + (1 - S_i)\gamma r_i \tag{17}$$

for $i = 1, 2$. $\tau_{NMDA} = 100\,\text{ms}$ and $\gamma = 0.641$ were the synaptic time constant and saturation factor for NMDA-R. $r_1$, $r_2$ were the firing rates of the two populations, and were quasi-statically computed from the transfer function based on the total input currents $I_1$, $I_2$ (*Figure 6D*). The input currents

$$I_1 = \alpha_1 S_1 + \alpha_2 S_2 + \beta_1 r_1 + \beta_2 r_2 + I_1^{ext} \tag{18}$$

$$I_2 = \alpha_1 S_2 + \alpha_2 S_1 + \beta_1 r_2 + \beta_2 r_1 + I_2^{ext} \tag{19}$$

arose from the NMDA-Rs of the same population (e.g. $\alpha_1 S_1$ in *Equation 18*) and competing population (e.g. $\alpha_2 S_2$ in *Equation 18*), the AMPA-Rs of the same population (e.g. $\beta_1 r_1$ in *Equation 18*) and competing population (e.g. $\beta_2 r_2$ in *Equation 18*), and external inputs (e.g. $I_1^{ext}$ in *Equation 18*).

GABA-Rs were also expressed in $\alpha_i$ and $\beta_i$ to account for lateral inhibition. Using change of variables $x_1 = \alpha_1 S_1 + \alpha_2 S_2 + I_1^{ext}$, $x_2 = \alpha_1 S_2 + \alpha_2 S_1 + I_2^{ext}$, the transfer function can be written as

$$r_1 = \frac{ax_1 - f(x_2) - b}{1 - exp[-d(ax_1 - f(x_2) - b)]} \tag{20}$$

$$r_2 = \frac{ax_2 - f(x_1) - b}{1 - exp[-d(ax_2 - f(x_1) - b)]} \tag{21}$$

where $a$, $b$, $d$ were constants that depended on $\beta_1$, and $f$ was a function of $x_i$ that depended on $\beta_2$. We omitted the expression of $\alpha_i, \beta_1, a, b, d, f, I_i^{ext}$ for the sake of simplicity, but please see *Wong and Wang, 2006* for details. The resulting transfer function (*Figure 6D*) is such that small input below a threshold generated no response, while very large input generated a linear response (note that *Figure 6D* shows $r_1$ as a function of $x_1$, with $x_2 = 0$). This resulted in an expansive non-linearity between the two limits, allowing strong inputs to drive the system more strongly than weak-inputs. Input streams with large variability, with higher chance to have both strong inputs and weak inputs, can thus leverage such asymmetry better than input streams with small variability, resulting in pro-variance bias (*Figure 6*).

$r_i$, as a (transfer) function of $I_i$, was sensitive to NMDA-R hypofunction due to $\alpha_1$, $\alpha_2$ in the first two terms in *Equations 18 and 19*. Through $\alpha_1$ and $\alpha_2$, NMDA-R hypofunction altered the transfer function and thus the expansive non-linearity, thus altering the pro-variance bias effect. In addition, parameter changes due to NMDA-R hypofunction ($\alpha_1$ and $\alpha_2$) will also alter the attractor dynamics of the circuit model, such that the perturbed circuit will have different dynamics and ranges of $S_1$ and $S_2$, resulting in a second indirect effect on the pro-variance bias effect.

The mean-field model completed 94,000 standard session 'Regular' trials, in the same manner as the circuit models. We only generated control circuits for the mean-field model. Predictions of perturbations from spiking circuit models generally held for the mean-field model. However, due to detailed distinctions in the dynamics of the spiking circuit model verses the mean-field model, perturbation-induced decision deficit arose from different mechanisms between the two sets of models (*Lam, 2017*). This complicated the translatability between the two sets of models, so we focused on the control circuit.

## Code and data availability

Stimuli generation and data analysis for the experiment were performed in MATLAB. The spiking circuit model was implemented using the Python-based Brian2 neural simulator (*Goodman and Brette, 2008*), with a simulations time step of 0.02ms. Further analyses for both experimental and model data were completed using custom-written Python and MATLAB codes. Data and analysis scripts to reproduce figures from the paper will be made publicly available for download from an online repository upon publication. Data has been uploaded to Dryad under the doi:10.5061/dryad.pnvx0k6k3. Code is available on GitHub at https://github.com/normanlam1217/CavanaghLam2020CodeRepository (copy archived at https://github.com/elifesciences-publications/CavanaghLam2020CodeRepository; *Lam, 2020*).

## Additional information

### Funding

| Funder | Grant reference number | Author |
| --- | --- | --- |
| National Institute of Mental Health | R01MH112746 | John D Murray |
| Wellcome | 098830/Z/12/Z | Laurence Tudor Hunt |
| Wellcome | 208789/Z/17/Z | Laurence Tudor Hunt |
| Brain and Behavior Research Foundation | | Laurence Tudor Hunt |
| National Institute for Health | | Laurence Tudor Hunt |

| | | |
|---|---|---|
| Research Oxford Health Bio-medical Research Centre | | |
| Middlesex Hospital Medical School General Charitable Trust | | Sean Edward Cavanagh |
| NSERC | PGSD2 - 502866 - 2017 | Norman H Lam |
| Wellcome | 096689/Z/11/Z | Steven Wayne Kennerley |

The funders had no role in study design, data collection and interpretation, or the decision to submit the work for publication.

### Author contributions

Sean Edward Cavanagh, Conceptualization, Data curation, Software, Formal analysis, Investigation, Visualization, Methodology, Writing - original draft, Writing - review and editing; Norman H Lam, Conceptualization, Data curation, Software, Formal analysis, Investigation, Visualization, Methodology, Writing - review and editing; John D Murray, Laurence Tudor Hunt, Conceptualization, Resources, Supervision, Funding acquisition, Writing - review and editing; Steven Wayne Kennerley, Conceptualization, Resources, Supervision, Funding acquisition, Project administration, Writing - review and editing

### Author ORCIDs

Sean Edward Cavanagh  https://orcid.org/0000-0001-9275-2725
Norman H Lam  https://orcid.org/0000-0001-5817-6680
John D Murray  https://orcid.org/0000-0003-4115-8181
Laurence Tudor Hunt  http://orcid.org/0000-0002-8393-8533
Steven Wayne Kennerley  https://orcid.org/0000-0002-5696-7507

### Ethics

Animal experimentation: All experimental procedures were approved by the UCL Local Ethical Procedures Committee and the UK Home Office (PPL Number 70/8842), and carried out in accordance with the UK Animals (Scientific Procedures) Act.

### Decision letter and Author response

Decision letter https://doi.org/10.7554/eLife.53664.sa1
Author response https://doi.org/10.7554/eLife.53664.sa2

## Additional files

### Supplementary files

• Supplementary file 1. Difference in log-likelihood of Full regression model (mean, SD, max, min, first, last of evidence values; *Equation 6* in Materials and methods) vs reduced model, for each monkey and the circuit model. Log-likelihood values were calculated using a cross-validation procedure (see Materials and methods). Column label refers to the removed regressor. Positive values indicate the full regression model performs better. Values depend on the number of completed trials, which differed both between subjects and the circuit model. For both monkeys and the circuit model, mean evidence is clearly the most important driver of choice behaviour, followed by first and last evidence samples which reflects the primacy bias. Finally, evidence standard deviation (SD) has a stronger effect than maximum and minimum evidence samples (Max and Min).

• Supplementary file 2. Difference in log-likelihood of regression models including either evidence standard deviation (SD) or both maximum and minimum evidence (Max and Min) as regressors, for each monkey and the circuit model. Log-likelihood values were calculated using a cross-validation procedure (see Materials and methods). Column label refers to the regressors additional to either SD or Max and Min. Positive values indicate the regression model with SD performs better than that with Max and Min. Values depend on the number of completed trials, which differed both between

subjects and the circuit model. Regardless of whether first and last evidence sample regressors are included, the models with standard deviation of evidence have higher log-likelihoods than the models with maximum and minimum evidence samples, indicating a better explanation of the data by standard deviation than by maximum and minimum evidence samples.

• Supplementary file 3. Increase in log-likelihood of various regression models (regressors in column labels) due to inclusion of evidence standard deviation as a regressor, for each monkey and the circuit model. Log-likelihood values were calculated using a cross-validation procedure (see Materials and methods). Values depend on the number of completed trials, which differed both between subjects and the circuit model. Positive values across the table indicates the evidence standard deviation regressor robustly improves model performance for all models examined.

• Supplementary file 4. Difference in log-likelihood of regression models including either evidence standard deviation (SD) or both maximum and minimum evidence (Max and Min) as regressors, for each monkey with saline or ketamine injection. Log-likelihood values were calculated using a cross-validation procedure (see Materials and methods). Column label refers to the regressors additional to either SD or Max and Min. Positive values indicate the regression model with SD performs better than that with Max and Min. Values depend on the number of completed trials, which differed across conditions. Regardless of whether first and last evidence sample regressors are included, the models with standard deviation of evidence have higher log-likelihoods than the models with maximum and minimum evidence samples, indicating a better explanation of the data by standard deviation than by maximum and minimum evidence samples. In particular, under ketamine injection, monkeys did not switch their strategy to primarily use maximum and minimum evidence samples (over standard deviation of evidence) to guide their choice.

• Transparent reporting form

## Data availability

Stimuli generation and data analysis for the experiment were performed in MATLAB. The spiking circuit model was implemented using the Python-based Brian2 neural simulator, with a simulations time step of 0.02ms. Further analyses for both experimental and model data were completed using custom-written Python and MATLAB codes. Data has been uploaded to Dryad under the doi:10.5061/dryad.pnvx0k6k3. Code is available on GitHub at https://github.com/normanlam1217/CavanaghLam2020CodeRepository (copy archived at https://github.com/elifesciences-publications/CavanaghLam2020CodeRepository).

The following dataset was generated:

| Author(s) | Year | Dataset title | Dataset URL | Database and Identifier |
|---|---|---|---|---|
| Cavanagh SE, Lam NH, Murray JD, Hunt LT, Kennerley SW | 2020 | Data from: A circuit mechanism for decision making biases and NMDA receptor hypofunction | https://doi.org/10.5061/dryad.pnvx0k6k3 | Dryad Digital Repository, 10.5061/dryad.pnvx0k6k3 |

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
