## [Decision Letter]

**Acceptance summary:**

This study uses a combination of neural circuit modeling with pharmacological intervention and behavioral psychophysics in monkeys to dissect the mechanisms of decision-making. It implicates the N-methyl-aspartate (NMDA) receptor in the accumulation of decision evidence, linking NMDA-mediated recurrent excitation of pyramidal neurons to a well-known behavioral phenomenon: a bias to choose options exhibiting larger variations in value. The approach opens up new perspectives for the mechanistic assessment of decision computations in the brain.

**Decision letter after peer review:**

Thank you for submitting your article "A circuit mechanism for decision making irrationalities and NMDA-R hypofunction: behaviour, modelling and pharmacology" for consideration by *eLife*. Your article has been reviewed by three peer reviewers, and the evaluation has been overseen by Tobias Donner as Reviewing Editor and Michael Frank as the Senior Editor. The following individual involved in review of your submission has agreed to reveal their identity: Konstantinos Tsetsos (Reviewer #1); Valentin Wyart (Reviewer #2).

The reviewers have discussed the reviews with one another and the Reviewing Editor has drafted this decision to help you prepare a revised submission.

While editors and reviewers found your work interesting in principle, all reviewers raised some substantial concerns that would need to be addressed before we can reach a final decision on your paper. The essential revisions are listed below. Indeed, it seems possible that the results of these requested analyses will require a substantial toning-down of several of your claims pertaining to E/I balance, in a way that could undermine the specificity of conclusions and the suitability of your paper for *eLife*. Even so, we agreed to give you the chance to address the concerns, for which two months should be a realistic time frame.

Summary:

This manuscript reports a computational and pharmacological study in monkeys, into a question of interest to a broad research community: The role of the NMDA receptor in evidence accumulation and decision-making. The authors used a protocol developed and tested in humans by Tsetsos and colleagues, in which subjects compare the average length of two sequences of visual bar stimuli. The monkeys exhibit a so-called “pro-variance bias” (PVB) toward choosing the more variable stream, although the monkey behavior differs from humans in other aspects (see below). The authors show that a neural spiking circuit model of bounded evidence accumulation shows a similar PVB, and that a lowered E/I ratio simultaneously decreases accuracy and increases PVB. Finally, they report that intramuscular injection of ketamine transiently decreases accuracy and increases PVB, as predicted by a lowered E/I ratio. The authors interpret their findings in the context of the previous work on PVB as well as pseudo-psychotic effects of ketamine in human subjects.

Essential revisions:

1) Specificity of the pharmacological claim within the circuit model.

You should show that the ketamine behavioural effects are robustly obtained under the lowered E/I hypothesis (e.g. for various magnitudes of E/I reduction ) and, crucially, incompatible with a) sensory deficit, b) elevated E/I, c) concurrent changes in both NMDA receptors. Practically, this means the following.

a) For each hypothesis, model predictions should be shown by varying the relevant model parameter(s) gradually within a range.

b) The similarity between model predictions and behavioural data should always be quantified using a goodness of fit metric (currently this is done by eye balling). Please should focus on perturbations who provide a good quantitative fit to the data.

c) Perturbations appear to be implemented in the same fashion as in Lam et al., 2017. There, the authors also changed other parameters besides the relevant synaptic weights, in order to maintain stability in the model dynamics. It is not clear if and how these extra changes could be pharmacologically induced by ketamine. Please clarify this aspect and derive predictions when stability adjustments are not performed.

2) Effect of drug on lapses.

Please test for a ketamine effect (sedation) on lapse rates. The psychometric functions under ketamine indicate a large change in the lapse rate which is currently not taken into account. All descriptive analyses (logistic regressions) and model simulations should take into account lapse rates. Can an increase in lapse rates explain away the changes in the PVB effect, psychometric curves, and kernels?

3) Validity of circuit model.

Currently, the circuit model is presented as a black box. You devote a couple of sentences in describing how the expansive non-linearities in the F-I curve give rise to the pro-variance effect. This part is not very well developed. One way to test whether indeed the non-linearities are crucial in the pro-variance effect the monkeys show, is to separately analyse trials with "high" (total sum of both streams high) vs. "low" (total sum of both streams low) evidence and see if the PVB effect changes. Or add the total sum as a regressor and compare the regression weights in the model and in the data. In addition to non-linearities in FI curves, the attractor dynamics of the circuit model may (or may not) promote the PVB effect. Are these dynamics even necessary to produce the pro-variance effect in the model? And is there any link between signatures of attractor dynamics (e.g. kernel shapes) and the PVB effect in the data? If dynamics were redundant in the model, would this undermine the claim that the PVB can be diagnostic of E-I balance?

This relates to the question concerning the way the PVB is quantified: in the model, how can the PVB index change even if the F-I non-linearity remains unchanged? It thus seems that the PVB index is sensitive to the overall signal-to-noise associated with the model and it is not a pure marker of the pro-variance propensity.

Please clarify what the PVB index stands for.

4) Results for both task framings.

Please present separate results for the two framings, i.e. "select higher" and "select lower" trials, which is interesting from an empirical viewpoint. Also: Have you mis-labelled the "high-variance correct" and "low-variance correct" trials in the "select the lower" conditions? (If not, then the quantification of the PVB may be wrong.)

5) Generalizability of findings to humans.

Reviewers raised doubts about the suggested analogy of monkey and human performance, and the underlying computations: Showing that both humans and monkeys have a PVB is not sufficient to establish a cross-species link. In the human work by Tsetsos et al. (PNAS, 2012, 2016), the temporal weighting of evidence on choice exhibits recency, in sharp contrast to the primacy found here in monkeys. What does this imply in terms of the relationship at a mechanistic level? This point needs to be discussed.

6) Link to schizophrenia.

Reviewers remarked that the link to schizophrenia is very loose: no patients are tested and overall behavioral signatures are different even from healthy human subjects (see point 3). Reviewers agreed that this point should at least be toned down substantially or dropped altogether. This tentative link could be brought up as speculation in Discussion, but not used as the basis for setting up the study.

7) Discuss limitations of pharmacological protocol.

a) The physiological effects of ketamine on cortical circuits remain speculative. The drug is unlikely to have the single, simple effect, as assumed in the model. This should be acknowledged in Discussion. Also, what happens in the model when NMDA hypofunction is implement in both neuron types?

b) The use of an intramuscular injection of ketamine at 0.5 mg/kg (about an order of magnitude stronger than what would be used in humans) produces a massive transient effect on task behavior, which has potential important drawbacks. First, the effect is massive, with decision accuracy dropping from about 85% correct to less than 60% correct after 5 minutes, followed by a sustained recovery over the next 30 minutes. This effect of ketamine is so strong that it is hard to know whether it is truly NMDA receptor hypofunction that produces the behavioral deficit, or task disengagement due to the substantial decrease in reward delivery (for example). The time window chosen for the analysis is also strongly non-stationary, and it is difficult to assess how much an average taken over this window is truly an accurate depiction of a common behavioral deficit throughout this time period (where accuracy goes from 60% correct to 80% correct). Again, the presence of possible attentional lapses should be accounted for (and reported in the manuscript) in all model fits and analyses, given the strength of ketamine-induced deficits triggered by this pharmacological protocol. We realize that this aspect of the study cannot be changed at this point, but it should be acknowledged as an important limitation.

[Editors' note: further revisions were suggested prior to acceptance, as described below.]

Thank you for resubmitting your article "A circuit mechanism for decision making biases and NMDA receptor hypofunction" for consideration by *eLife*. Your revised article has been reviewed by 2 peer reviewers, and the evaluation has been overseen by a Reviewing Editor and Michael Frank as the Senior Editor. The following individuals involved in review of your submission have agreed to reveal their identity: Konstantinos Tsetsos (Reviewer #1); Valentin Wyart (Reviewer #2).

The reviewers have discussed the reviews with one another and the Reviewing Editor has drafted this decision to help you prepare a revised submission.

We would like to draw your attention to changes in our revision policy that we have made in response to COVID-19 (https://elifesciences.org/articles/57162). Specifically, when editors judge that a submitted work as a whole belongs in *eLife* but that some conclusions require a modest amount of new analyses, as they do with your paper, we are asking that the manuscript be revised to either limit claims to those supported by data in hand, or to explicitly state that the relevant conclusions require additional supporting analyses.

Our expectation is that the authors will eventually carry out the additional analyses and report on how they affect the relevant conclusions either in a preprint on bioRxiv or medRxiv, or if appropriate, as a Research Advance in *eLife*, either of which would be linked to the original paper.

Summary:

The authors have provided an extensive response to the reviewers' comments based on several additional analyses of their data; they have successfully addressed a large subset of the comments. Specifically, they have performed several additional analyses to (i) test alternative hypotheses as well as the robustness of the favored hypothesis, (ii) examine lapses under ketamine, (iii) unpack the workings of the circuit model, and (iv) examine the frequent-winner effect in the data so they can assess the generalizability of this study to humans. We acknowledge that all these analyses have led to a significant improvement. Nevertheless, we remain uncertain about the validity of the overall conclusion, that ketamine induces NMDA-R hypo-function in excitatory neurons, and that this effect is behaviorally manifested as an increase in a pro-variance bias.

1) Motivate modeling approach.

Given that you opted not to fit the model (which would be done with the meanfield reduction), or tune its parameters so that it matches the above behavioral patterns, we believe you should unpack the reasoning underlying this particular modeling approach.

2) Plot model predictions along with data.

As we pointed in our first review there seem to be some discrepancies between the data and the model, which we remain concerned about:

i) Ketamine data asymptote at a lower than 100% level. The lapse rates are still not plugged in the circuit model so as to bring the model predictions closer to the data.

ii) The control kernel in Figure 7I and the monkey kernels in Figure 8C look different. In the model, there is a primacy pattern (except for the first item) but in the data we see a flat/ U-shaped pattern. Plotting those together could reveal the degree of discrepancy.

iii) In the control condition, the psychometric functions in Figure 7B and in Figure 8B look different (for example in terms of convergence of the light and dark coloured lines). The elevated E/I plot in Figure 7D appears to be closer to the saline psychometric curve.

Such discrepancies, if true, matter: if the "baseline" model does not capture behavior in the control condition well, we cannot be confident about the validity of the subsequent perturbations performed to emulate the ketamine effect. To allow for better assessing the match, we strongly encourage you to always plot model predictions with the data.

Ideally, you would also assess the goodness of fit using maximum likelihood. (A certain parametrization could exhibit similarity with the data in terms of the logistic regression weights (PVB) but at the same time it can miss largely on capturing the psychometric function.) We believe this would be straightforward, given the simulations you have already performed, but leave the decision to you, whether to not to do this.

We realize that this point was not explicitly raised in the previous round. Then, reviewers had asked for a quantification of the goodness of fit. The approach you chose (logistic regression) is specific to the PVB index (not applied to psychometric functions and kernels) and did not fully convince reviewers.

3) Assess effects of concurrent NMDA-blockade on E and I neurons.

You establish that E/I increase reduces the PVB index while E/I decrease has the opposite effect. However, you have not examined the effect of concurrent changes of NMDA-Rs of both, E and I cells, which we had suggested to do. Please comment on the fact that concurrent changes could mimic the effect of E-E reduction (Figure 7—figure supplement 2: moving up diagonally the purple point would result in equivalent behavior). Unless there is strong support in favor of the selective NMDA change over the concurrent change (assessed via maximum likelihood), the conclusions should be reframed.

4) Add a lapse rate downstream from circuit model.

You have now assessed lapse rates in your analysis, but reviewers remarked that you do not report the best-fitting lapse rates. This makes it impossible to judge just how much lapses contribute to the decrease in task performance in the initial period following ketamine injection (which is included in all analyses). We are concerned that this massive performance drop under ketamine is not only triggered by aPVB, but also (perhaps largely) by an increase in lapses and a decrease in evidence sensitivity.

We would expect a lapse mechanism to be in play in the circuit model when emulating the ketamine effect. You could use the fraction of lapses best fitted to psychometric curves (which clearly do not saturate at p(correct) = 1) for the circuit model simulations. It seems conceivable that allowing the circuit model to lapse will reduce the weight applied on the mean evidence.

5) Different quantification of pro-variance bias.

We do not understand the motivation for compressing sensitivity to mean and to variance into a single PVB index. Our reading is that the pro-variance effect, quantified as a higher probability of choosing a more variable stream (see Tsetsos et al., 2012), can just be directly mapped onto the variance regressor. Combining the weights into a PVB index and framing the general discussion around this index seems unnecessary. The main behavioral result of ketamine can be parsimoniously summarized as a reduced sensitivity to the mean evidence. Relatedly, please discuss if and how the ketamine-induced increase in the PVB effect, the way you quantified it, rides over a strong decrease of the sensitivity to mean evidence under ketamine.

It does seem to be the case that sensitivity to variance remains statistically indistinguishable between saline and ketamine (if anything it is slightly reduced). The E/I increase model consistently predicts that the variance regressor is reduced. This is not the case with the E/I decrease model, which occasionally predicts increases in the sensitivity to the variance (see yellow grids in Figure 7—figure supplement 2). This feature of the E/I decrease model should be discussed, as it seems to undermine the statement that the E/I perturbation produces robust predictions regardless of perturbation magnitude (i.e. depending on the strength of E/I reduction the model can produce a decrease or increase on variance sensitivity, and the relationship is non-monotonic). Overall, we believe that combining sensitivity to mean and variance obscures the interpretation of the data and model predictions.

Again, we realize that this point appears to be new. But reviewers feel they could not really have a strong case regarding this metric without seeing the more detailed model predictions (in a 2-d grid) that you have presented in your revision.

---

## [Author Response]

Essential revisions:1) Specificity of the pharmacological claim within the circuit model.You should show that the ketamine behavioural effects are robustly obtained under the lowered E/I hypothesis (e.g. for various magnitudes of E/I reduction) and, crucially, incompatible with a) sensory deficit, b) elevated E/I, c) concurrent changes in both NMDA receptors. Practically, this means the following.a) For each hypothesis, model predictions should be shown by varying the relevant model parameter(s) gradually within a range.b) The similarity between model predictions and behavioural data should always be quantified using a goodness of fit metric (currently this is done by eye balling). Please should focus on perturbations who provide a good quantitative fit to the data.c) Perturbations appear to be implemented in the same fashion as in Lam et al., 2017. There, the authors also changed other parameters besides the relevant synaptic weights, in order to maintain stability in the model dynamics. It is not clear if and how these extra changes could be pharmacologically induced by ketamine. Please clarify this aspect and derive predictions when stability adjustments are not performed.

We thank the reviewers for this comment. We agree it is important to demonstrate the robustness of our model predictions. We have therefore included a 2-dimensional parameter scan with simultaneous NMDA-R hypofunction on excitatory (which lowers E/I) and inhibitory (which elevates E/I) neurons in the circuit model. We have also included a 1-dimensional parameter scan of the sensory deficit perturbation strength. Crucially, these parameter scans demonstrate robust effects by perturbations on the PVB index and the majority of the regression coefficients, in the three directions of lowered E/I, elevated E/I, and sensory deficit (new Figure 7—figure supplements 2, 3, and 4 ). In particular, PVB index is consistently increased by lowered E/I, decreased by elevated E/I, and unaltered by sensory deficit. Extremely strong sensory deficit resulted in an increase in PVB index, but this effect occurred at the limit where the model can barely perform the task (Figure 7—figure supplement 4), with a psychometric function qualitatively different from the monkey behaviour under ketamine (Figure 8—figure supplement 6).

To address comment 1b, we need to define an appropriate measure to quantify the degree to which the perturbation in the model alters decision-making behaviour in a similar manner as does ketamine in the monkeys. Importantly, the control parameters of the biophysically-based spiking circuit model were not at all fit to the monkey’s baseline behaviour (which is typical for spiking circuit modelling), and instead were the same as in Lam et al., 2017. Despite differences between model and monkey in control psychometric performance, we can quantify whether a perturbation produces a similar *change* in performance. The same could be applied for the two monkeys – despite baseline differences, does ketamine alter behaviour similarly between them?

Here, we focused on two key aspects of behavioural alteration: the relative changes in the (i) evidence mean and (ii) evidence standard deviation regression weights. We then quantify the comparison between two sets of change (e.g., model to monkey, or between two monkeys) as the cosine similarity (CS) of the two vectors composed of these relative changes (Figure 8—figure supplement 4A). Applying this measure to compare between the two monkeys, we find CS = 0.94, corresponding to an angle of 20.1 degrees, which shows the consistency of ketamine effects between the monkeys.

We applied this analysis to quantify the similarity between a monkey’s behaviour change under ketamine and the model under a range of parameter perturbations (2D sweeps of NMDAR hypofunction, and sensory deficit) (Figure 8—figure supplement 4B-I). These analyses found that the lowered E/I perturbation robustly yielded a similar performance change as measured in the monkeys under ketamine, with higher CS values than elevated E/I or sensory deficit perturbations. Specifically, the 1D sweep of lowered E/I yielded maximum CS values of 0.9972 and 0.9968 for Monkeys A and H, respectively (comparable to the between-monkey CS of 0.9391). These results were replicated by model comparison analysis using Euclidean distance, as a metric which also accounts for the magnitude of the vectors (Figure 8—figure supplement 5).

It is important to note that our modelling results support the hypothesis of lowered E/I in decision making circuits contributing to the pro-variance effect, but cannot exclude possible contributions from sensory deficits (which will not alter the pro-variance bias in our model). For the same reason, we did not consider a 2-dimensional parameter scan with both lowered E/I and sensory deficit perturbations, as no dissociable predictions can be inferred from that analysis.

The cosine similarity and Euclidean distance analyses, motivated by comment 1b, informed us that a moderately weaker perturbation of lowered E/I (by ~25%) yielded a better fit than the perturbation strength in our original submission, to the pattern of behavioural alteration observed under ketamine (Figure 8—figure supplement 4D,G and 5D,G). We have therefore updated the main Figure 7 with a lowered E/I perturbation strength that is a better fit by this measure, along with other perturbations to match the reduction in evidence mean regression weight.

Regarding comment 1c, we would like to clarify that the control circuit model in the current study is identical to that in Lam et al., 2017. The only parameter which is different is μ, which scales the input current as a function of the visual stimulus; given the different task paradigms, we believe it is reasonable to retune μ to better match the observed experimental data. The control circuit model in both the current study and in Lam et al., 2017 are different from the model presented in Wang, 2002. As originally noted in Lam et al., 2017, adjustments were made to the Wang, 2002 parameters to have stability of baseline and memory states under a wider range of E/I perturbations. (We note that all of the same qualitative effects of altered E/I can be observed in the Wang, 2002 parameters, but within a smaller range of perturbation strengths.)

Importantly, in both the current study and in Lam et al., 2017, we considered the control circuit model as the default state, corresponding to no pharmacological E/I perturbation. Therefore, the adjustments to control parameters from Wang, 2002 to the present study are not part of the simulated effects of pharmacological perturbation. The simulated effect of the perturbation on the local circuit, corresponding to ketamine, is solely mediated by reducing the conductance of recurrent NMDA receptors.

For the reviewers’ convenience, we included additions to the manuscript in response to this comment. In response to comment 1a, we added the following text to the Results:

“While all circuit models were capable of performing the task (Figure 7B-E), the choice accuracy of each perturbed model was reduced when compared to the control model. […] Together, the circuit model thus provided the basis of dissociable prediction by E/I-balance perturbing pharmacological agents.”

We also added the details of parameter scans to test the robustness of model prediction

in the Materials and methods, in response to comments 1a and 1b:

“[…] For the exact parameters, the lowered E/I model reduced GE→E by 1.3125%, the elevated E/I model reduced GE→I by 2.625%, and the sensory deficit model had a sensory deficit of 20% (such that μ′ was reduced by 20%) (Figure 7, Figure 7—figure supplement 1). […] A higher cosine similarity (and lower Euclidean distance) meant the relative extent (and direction) of alteration, to the regression coefficients of mean evidence and evidence standard deviation, was more similar between the perturbations in the circuit model and the monkey data.”

In response to comment 1b, we added the following text to the Results:

“Additional observations further supported the lowered E/I hypothesis for the effect of ketamine on monkey choice behaviour. […] This shifting of the weights could reflect a sensory deficit, but given the results of the pro-variance analysis, collectively the behavioural effects of ketamine are most consistent with lowered E/I balance and weakened recurrent connections.”

2) Effect of drug on lapses.Please test for a ketamine effect (sedation) on lapse rates. The psychometric functions under ketamine indicate a large change in the lapse rate which is currently not taken into account. All descriptive analyses (logistic regressions) and model simulations should take into account lapse rates. Can an increase in lapse rates explain away the changes in the PVB effect, psychometric curves, and kernels?

Thank you for raising this important point. As the term lapse rate is slightly ambiguous, we will initially provide some clarification. Lapse rate may refer to the rate at which incomplete trials occur (i.e. due to the subject not responding, or breaking fixation). Alternatively, it may refer to the animal responding randomly, regardless of the trial difficulty, on a certain proportion of trials. Our response below will address both of these factors.

Firstly, in our initial submission, all incomplete trials (i.e. those where the animal did not commit to a choice, or broke fixation) were excluded from the analyses. The only trials included in the analyses were those where the animal completed a choice. Hence, any change in our accuracy measure (i.e. as in Figure 8A) relates specifically to changes in their actual choices, rather than task engagement. It is also important to stress that these “incomplete trials” occurred rarely, even when the animals were administered with ketamine:

**Author response image 1. sa2fig1:** Animals rarely fail to complete trials when administered with ketamine. The proportion of incomplete trials that occurred between 5 minutes and 30 minutes relative to drug administration. Errorbars indicate the standard error, each dot represents an individual session. Monkey A counterintuitively completed a higher proportion of trials when administered with ketamine, as this appeared to reduce his slightly stronger tendency to break central fixation early in order to directly view the stimuli. For Monkey H, the proportion of incomplete trials was relatively uninfluenced by ketamine.

The second type of lapsing, random responses, are an important consideration that our initial submission did not address. As the reviewers suggest, it is possible that an increase in these types of lapses could account for the animals’ reduction in accuracy when administered with ketamine. To address this point, we extended our existing logistic regression models to incorporate an extra parameter which could account for these lapses. The benefits of including this parameter were twofold:

1) To quantify the lapse rate

2) To control for lapsing, and isolate its effect from our other analyses (i.e. PVB index, kernels).

The updated models are listed below (description taken from the revised Materials and methods):

“To control for possible lapse effects induced by ketamine, where the animal responded randomly regardless of the trial difficulty, the behavioural models described above were extended to include an extra “lapse parameter”, Y_0_. […] Bootstrapping was used to generate error estimates for the parameters of these models (10,000 iterations). As our analyses demonstrate that the animals very rarely lapse when administered with saline, we did not deem it necessary to apply the lapsing models to the standard session experiment (i.e. Figures 2-6). ”

Crucially, our existing analyses of the ketamine data were not affected when controlling for lapses. It was clear that accounting for lapse rates did not explain away the changes in the PVB effect or the kernels. We have included these new results as a supplementary figure to the main Figure 8. See Figure 8—figure supplement 2.

For the reviewers’ convenience, we have also included Author response image 2 which compares the results from the original submission with the updated results utilising the lapsing model:

**Author response image 2. sa2fig2:** Incorporating a lapsing parameter does not greatly influence coefficients from the original logistic model for the PVB analysis. (A) The mean evidence regression coefficient under saline (blue) and ketamine (red) under logistic regression with (no hatches) or without (hatched) a lapse term, using Monkey A data. (B) Same as (A) but using Monkey H data instead. (C-D) Same as (A-B) but for the evidence standard deviation regression coefficient instead. (E-F) Same as (A-B) but for PVB index instead. Note that while both mean evidence and evidence standard deviation regression coefficients under ketamine injection vary with or without the lapse term, the PVB index is stable to the lapse term. All errorbars denote the 95% confidence interval generated through a bootstrap procedure.

As the reviewers implied, the subjects’ lapsing did increase with ketamine. Whilst we have robustly established this is not the cause of our behavioural effects, we felt this was an important point to include in the manuscript. We have therefore updated the main text in the Results section together with changes from comment 1b:

“To understand the nature of this deficit, we studied the effect of drug administration on the pro-variance bias (Figure 8B-F). […]This confirmed that the rise in PVB was an accurate description of a common behavioural deficit throughout the duration of ketamine administration.”

As mentioned in the Materials and methods, our analyses demonstrate that the animals very rarely lapse when administered with saline. As such, we did not deem it necessary to apply the lapsing models to the standard session experiment (i.e. Figures 2-6). With regards to the psychometric functions (e.g. Figures 8B-C), these have not been updated. This is because the three parameters in this model (Equation 2) are already sufficient to capture lapsing behaviour. Regardless, these psychometrics are purely illustrative and are not used in any of the statistical reporting.

3) Validity of circuit model.Currently, the circuit model is presented as a black box. You devote a couple of sentences in describing how the expansive non-linearities in the F-I curve give rise to the pro-variance effect. This part is not very well developed. One way to test whether indeed the non-linearities are crucial in the pro-variance effect the monkeys show, is to separately analyse trials with "high" (total sum of both streams high) vs. "low" (total sum of both streams low) evidence and see if the PVB effect changes. Or add the total sum as a regressor and compare the regression weights in the model and in the data. In addition to non-linearities in FI curves, the attractor dynamics of the circuit model may (or may not) promote the PVB effect. Are these dynamics even necessary to produce the pro-variance effect in the model? And is there any link between signatures of attractor dynamics (e.g. kernel shapes) and the PVB effect in the data? If dynamics were redundant in the model, would this undermine the claim that the PVB can be diagnostic of E-I balance?This relates to the question concerning the way the PVB is quantified: in the model, how can the PVB index change even if the F-I non-linearity remains unchanged? It thus seems that the PVB index is sensitive to the overall signal-to-noise associated with the model and it is not a pure marker of the pro-variance propensity.Please clarify what the PVB index stands for.

We thank the reviewers for raising these important issues, and for suggesting an interesting analysis which we now include. We agree the mechanism of the pro-variance effect from the decision making process could be further analysed and explained, especially regarding the expansive non-linearities in the F-I curve. We have now expanded on Results, Materials and methods, and Discussion, to discuss how the evidence integration process can generate a pro-variance effect. We also discussed the relation of this mechanism with attractor dynamics, the comparison of this mechanism with the selective integration model (Tsetsos et al, 2016), and how E/I balance disruption may change the F-I non-linearity in the mean-field model and thus impact the PVB index.

In particular, regarding the reviewers’ comment on how attractor dynamics may contribute to a pro-variance bias, we want to highlight that in recurrent circuit models, there is not a clean separation between attractor dynamics and the other factors impacting evidence integration, e.g. to disentangle contributions to PVB. This is in contrast to the Tsetsos et al, 2016 model, which has separable stages from nonlinear transformation of evidence, to the process of integrating that transformed evidence. Figure 6E-H illustrates that in the recurrent circuit, the temporal change of the systems state (S1, S2) depends on the current state (S1, S2) itself, exhibiting an attractor landscape. Furthermore, Figure 6D-H shows that this attractor landscape itself reconfigures dynamically as the stimulus input changes. In a sense, the “gain” of how stimulus impacts the state (i.e. how it is integrated) varies dynamically as a function of both stimulus and the stochastically evolving state of the system (see Materials and methods). This is why these factors cannot be disentangled. These points are now included in the Discussion. Nonetheless, we do agree that future theoretical analysis would be useful to help to link biophysical circuit models, reduced as nonlinear dynamical systems, to more tractable evidence accumulation models (e.g. selective integration). Such algorithmic models may allow us to unveil how various signatures of attractor dynamics are linked to the PVB effect, as raised by the reviewers. For instance, a short integration timescale demonstrated by elevated E/I circuits (Figure 7I) would prevent within-trial variabilities of the stimulus from being inferred, especially when only one or two bars are integrated.

Based on the suggestion for a new analysis, we tested for differential effects of “high” vs. “low” amounts of total evidence, in both the model and the monkeys (Figure 5—figure supplement 2). In the circuit model, trials with more total evidence more strongly drive the neurons to the near-linear regime of the F-I curve, and thus have a smaller PVB index than trials with less total evidence. Interestingly, the monkeys also demonstrated a consistent trend, though this effect did not achieve statistical significance. The temporal regression weights were also different between more vs. less total evidence, consistently between model and monkeys. The Results section is now expanded to discuss the support of the F-I non-linearity and more generally attractor dynamics from this analysis.

Finally, in relation to the question about how can the PVB index change even if the F-I non-linearity remains unchanged, we now include more details on our mean-field model in the Materials and methods section, in order to explain how E/I balance disruption may lead to changes in PVB index. The transfer function as a function of variables x1 and x2 (Equations 18, 19) is unchanged across the circuit models. However, x1 and x2 can be expressed in terms of underlying input currents, and the transfer function thus expressed as a function of the synaptic currents (I1 and I2) depends on NMDA-R mediated recurrent interactions. As a result, the effective transfer function *on the stimulus input* is actually altered by E/I perturbation (because E/I perturbation changes the recurrent contributions to synaptic currents). As such, NMDA-R hypofunction alters PVB index, both due to changes in NMDA-R coupling strengths (α1 and α2), and also from distinct dynamics and ranges of S1 and S2 as a result of different α1 and α2.

The updated texts are included below for the reviewers’ convenience.

In Results:

“To understand the origin of the pro-variance bias in the spiking circuit, we mathematically reduced the circuit model to a mean-field model (Figure 6A), which demonstrated similar decision-making behaviour to the spiking circuit (Figure 6B-C, Figure 6—figure supplement 1). […] In addition, distinct temporal weighting on stimuli were observed in both the circuit model and experimental data, for trials with more versus less total evidence (Figure 5—figure supplement 2D,H).”

In Materials and methods:

“The current spiking circuit model was mathematically reduced to a mean-field model, as outlined in (Niyogi and Wong-Lin, 2013), in the same manner as from (Wang, 2002) to (Wong and Wang, 2006). […] This complicated the translatability between the two sets of models, so we focused on the control circuit.”

In Discussion:

“The results from our spiking circuit modelling also provided a parsimonious explanation for the cause of the pro-variance bias within the evidence accumulation process. […] While other phenomenological models may also explain pro-variance bias, their link to our circuit model is similarly indirect, and were out of the scope of this study.”

4) Results for both task framings.Please present separate results for the two framings, i.e. "select higher" and "select lower" trials, which is interesting from an empirical viewpoint. Also: Have you mis-labelled the "high-variance correct" and "low-variance correct" trials in the "select the lower" conditions? (If not, then the quantification of the PVB may be wrong.)

Thank you for this suggestion. In response to this point, we have included three additional supplementary figures (Figure 2—figure supplement 1, Figure 3—figure supplement 2, Figure 4—figure supplement 2). It is clear from these figures that very similar results are attained for all analyses regardless of the task framing.

Unfortunately, we are slightly unclear what the reviewers meant with regards to the mislabelling of conditions. To clarify, the quantification of the PVB is determined by Equation 5:(5)ln⁡(PL1− PL)= β0 + β1(mean(L)−mean(R))+ β2 (std(L)−std(R))where P_L_ refers to the probability of choosing the left option, β0 is a bias term, β1 reflects the influence of evidence mean, and β2 reflects the influence of standard deviation of evidence (evidence variability). Author response table 1 outlines how this relates to the bar heights in each of the conditions:

**Author response table 1. resptable1:** 

Condition	Variable	Description
“Select Higher”	mean(L)	Average height of the 8 bars on the left side of the screen
“Select Higher”	mean(R)	Average height of the 8 bars on the right side of the screen
“Select Higher”	std(L)	Standard deviation of the heights of the 8 bars on the left side of the screen
“Select Higher”	std(R)	Standard deviation of the heights of the 8 bars on the right side of the screen
“Select Lower”	mean(L)	Average of (100 – Bar Height) for the 8 stimuli on the left side of the screen
“Select Lower”	mean(R)	Average of (100 – Bar Height) for the 8 stimuli on the right side of the screen
“Select Lower”	std(L)	Standard deviation of (100 – Bar Height) for the 8 stimuli on the left side of the screen
“Select Lower”	std(R)	Standard deviation of (100 – Bar Height) for the 8 stimuli on the right side of the screen

In the main paper (i.e. Figure 4D), the analysis is not calculated separately for “select higher” and “select lower” conditions. Furthermore, it does not depend on whether the trial is labelled as “high-variance correct” or “low-variance correct”. The purpose of these labels was only for visualisation as part of the psychometric plots (Figure 4C).

We believe some of this confusion may be resulting from the terminology we are using. To address this, we have updated references to “select higher” and “select lower” to “select taller” and “select shorter”. For example,

“Subjects were presented with two series of eight bars (evidence samples), one on either side of central fixation. Their task was to decide which evidence stream had the taller/shorter average bar height, and indicate their choice contingent on a contextual cue shown at the start of the trial.”

“Subjects had previously learned that two of these cues instructed to choose the side with the taller average bar-height (“ChooseTallTrial”), and the other two instructed to choose the side with the shorter average bar-height (“ChooseShortTrial”).”

We have also added the following sentences to the Materials and methods section to add some clarity with how this ties in with the illustrative psychometric plots of the pro-variance:

“To illustrate the effect of pro-variance bias, we also fitted a three-parameter psychometric function to the subjects’ probability to choose the higher SD option (PHSD) in the “Regular” trials, as a function of the difference in mean evidence in favour of the higher SD option on each trial (xHSD[…] On “ChooseShortTrials”, the mean evidence in favour of the higher SD option was calculated by subtracting (100 – mean bar height of the lower SD option) from (100 – mean bar height of the higher SD option).”

To clarify, it is not necessary to split the results for the two framings for the circuit model data. This is because the inputs to the circuit model are the transformed evidence values (i.e. bar height on “Select Higher” trials; 100 – bar height on “Select Lower trials”). Therefore, the circuit model will not show any difference in results between the two task framings.

5) Generalizability of findings to humans.Reviewers raised doubts about the suggested analogy of monkey and human performance, and the underlying computations: Showing that both humans and monkeys have a PVB is not sufficient to establish a cross-species link. In the human work by Tsetsos et al. (PNAS, 2012, 2016), the temporal weighting of evidence on choice exhibits recency, in sharp contrast to the primacy found here in monkeys. What does this imply in terms of the relationship at a mechanistic level? This point needs to be discussed.

Thanks for raising this point. We agree that there are differences between the primacy bias found in our paradigm and the recency bias found in the previous Tsetsos papers. We now discuss this point in the Discussion:

“Crucially, our circuit model generated dissociable predictions for the effects of NMDA-R hypofunction on the pro-variance bias (PVB) index that were tested by follow-up ketamine experiments. […] A stronger test will be to record neurophysiological data while monkeys are performing our task; this would help to distinguish between the “selective integration” hypothesis and the cortical circuit mechanism proposed here.”

6) Link to schizophrenia.Reviewers remarked that the link to schizophrenia is very loose: no patients are tested and overall behavioral signatures are different even from healthy human subjects (see point 3). Reviewers agreed that this point should at least be toned down substantially or dropped altogether. This tentative link could be brought up as speculation in Discussion, but not used as the basis for setting up the study.

Thanks for this comment. We agree that our previous version focussed too heavily on the potential link to schizophrenia, and that it is indeed unreasonable for us to do this without including data from patients or human volunteers. As such, we have extensively rewritten the Abstract, significance statement, and Introduction to tone them down substantially. In particular, we have removed most of the references to schizophrenia that were found throughout the previous version.

On the other hand, we think that it is reasonable to discuss the relationship between NMDA-receptor hypofunction and its effects on cognition and behaviour (we are directly manipulating/measuring these in the present study). We also feel that it is important to motivate this with an initial reference to the (vast) literature on NMDA-R antagonism via ketamine administration as an acute model of schizophrenia in humans. This was, after all, one of the main motivating factors for wanting to characterise the effects of ketamine in the present task.

We have therefore kept an initial reference to this relationship at the beginning of the Introduction, and then in the rest of the Introduction have limited our discussion to those of mechanisms of action of ketamine and NMDA-R hypofunction, rather than schizophrenia. We hope that the reviewers find this to be a reasonable compromise.

7) Discuss limitations of pharmacological protocol.a) The physiological effects of ketamine on cortical circuits remain speculative. The drug is unlikely to have the single, simple effect, as assumed in the model. This should be acknowledged in Discussion. Also, what happens in the model when NMDA hypofunction is implement in both neuron types?

We thank the reviewers for this excellent point and agree that we should address the complex effect of ketamine on the brain. We now discuss that point in Discussion (see below). Regarding the effects when NMDA hypofunction is implemented in both neuron types, this is covered in our response to major comment 1.

“Our pharmacological intervention experimentally verified the significance of NMDA-R function for decision-making. […] Finally, receptors of other brain areas might also be altered by intramuscular ketamine injection, which is beyond the scope of the microcircuit model in this study.”

b) The use of an intramuscular injection of ketamine at 0.5 mg/kg (about an order of magnitude stronger than what would be used in humans) produces a massive transient effect on task behavior, which has potential important drawbacks. First, the effect is massive, with decision accuracy dropping from about 85% correct to less than 60% correct after 5 minutes, followed by a sustained recovery over the next 30 minutes. This effect of ketamine is so strong that it is hard to know whether it is truly NMDA receptor hypofunction that produces the behavioral deficit, or task disengagement due to the substantial decrease in reward delivery (for example). The time window chosen for the analysis is also strongly non-stationary, and it is difficult to assess how much an average taken over this window is truly an accurate depiction of a common behavioral deficit throughout this time period (where accuracy goes from 60% correct to 80% correct). Again, the presence of possible attentional lapses should be accounted for (and reported in the manuscript) in all model fits and analyses, given the strength of ketamine-induced deficits triggered by this pharmacological protocol. We realize that this aspect of the study cannot be changed at this point, but it should be acknowledged as an important limitation.

Thank you for this comment. We have structured our response to first address the reviewers’ concerns regarding the drug dose and administration route. Then we address the reviewers’ point about task disengagement. Finally, we address the point regarding the analysis time window. We have previously addressed accounting for attentional lapses in our response to reviewer comment 2.

i) Firstly, we acknowledge that an intravenous infusion approach would have advantages over intramuscular injections. However, this was not possible because it was not within the remit of the ethical approval granted by the local ethical procedures committee and UK Home Office. Despite this, it is important to stress that intramuscular injections of ketamine at around 0.5mg/kg has been the standard approach used in several previous non-human primate studies (see Author response table 2). We are not aware of any non-human primate cognitive neuroscience studies that have used an infusion approach.

**Author response table 2. resptable2:** 

**Authors**	**Journal**	**Intramuscular Ketamine Doses Used**
(M. Wang, Yang et al., 2013)	Neuron	0.5-1.5 mg/kg
(Blackman, Macdonald et al., 2013)	Neuropsychopharmacology	0.32–0.57 mg/kg
(Ma, Skoblenick et al., 2015)	Journal of Neuroscience	0.4 mg/kg
(Ma, Skoblenick et al., 2018)	Journal of Neuroscience	0.4 – 0.7 mg/kg
(Shen, Kalwarowsky et al., 2010)	Journal of Neuroscience	0.25 – 1 mg/kg
(K. J. Skoblenick, Womelsdorf et al., 2016)	Cerebral Cortex	0.4 mg/kg
(K. Skoblenick and Everling, 2014)	Journal of Cognitive Neuroscience	0.4 mg/kg
(K. Skoblenick and Everling, 2012)	Journal of Neuroscience	0.4 – 0.8 mg/kg
(Taffe, Davis et al., 2002)	Psychopharmacology	0.3- 1.7 mg/kg
(Condy, Wattiez et al., 2005)	Biological Psychiatry	0.2 – 1.2 mg/kg
(Stoet and Snyder, 2006)	Neuropsychopharmacology	0.07 – 1 mg/kg

Secondly, as stated in our original submission, we extensively piloted different doses ranging from 0.1 – 1.0 mg/kg before data collection began. 0.5mg/kg was chosen as it was consistently inducing a performance deficit, while not causing significant task disengagement.

Finally, with regards to the chosen dose, we respectfully disagree that it is an order of magnitude stronger than that used in humans. Although it is slightly difficult to compare with relevant human studies as the vast majority of these have used infusion approaches, we will consider one such protocol (Anticevic, Gancsos et al., 2012; Corlett, Honey et al., 2006). In these studies, the authors gave an initial *intravenous* bolus of 0.23 mg/kg over 1 minute, followed by a subsequent continuous target controlled infusion (0.58 mg/kg over 1 h; plasma target, 200 ng/mL). This dose is relatively similar to what could be expected shortly after a 0.5mg/kg intramuscular injection. Furthermore, in the most relevant intramuscular study we could find, (Ghoneim, Hinrichs et al., 1985) did use intramuscular injections of ketamine to study its cognitive effects in humans. The dose they used was 0.25 – 0.5 mg/kg.

ii) With regards to task disengagement, we did not find evidence of a significant increase in incomplete trials (see response to reviewer comment 2, Author response image 1). Although we did find the animals lapse more frequently when administered ketamine, our behavioural effects were still present when controlling for this (see response to reviewer comment 2).

iii) The reviewers make a good point with regards to the analysis time window. Firstly, a similar approach of averaging across all trials after an intramuscular injection has been used in previous non-human primate studies (Blackman, Macdonald et al., 2013; Ma, Skoblenick et al., 2018; Ma, Skoblenick et al., 2015; K. Skoblenick and Everling, 2012, 2014; K. J. Skoblenick, Womelsdorf et al., 2016; M. Wang, Yang et al., 2013). However, we agree that it would be beneficial to investigate this further. To determine the time course of ketamine’s influence on the PVB index, we ran a sliding regression analysis:

“We later revisited the time course of drug effects by running our regression analyses at each of the binned windows described above (Figure 8—figure supplement 3[…] The true cluster size was significant at the p < 0.05 level if the true cluster length exceeded the 95th percentile of the null distribution.”

This reveals that the increase in PVB index is present when data from individual time periods are analysed. We believe this should allay the reviewers’ concern that the increase in PVB index is not a common behavioural deficit throughout this time period.

iv) The presence of possible attentional lapses has been accounted in all the drug day analyses. This was covered in our response to reviewer comment 2 above.

[Editors' note: further revisions were suggested prior to acceptance, as described below.]

Summary:The authors have provided an extensive response to the reviewers' comments based on several additional analyses of their data; they have successfully addressed a large subset of the comments. Specifically, they have performed several additional analyses to (i) test alternative hypotheses as well as the robustness of the favored hypothesis, (ii) examine lapses under ketamine, (iii) unpack the workings of the circuit model, and (iv) examine the frequent-winner effect in the data so they can assess the generalizability of this study to humans. We acknowledge that all these analyses have led to a significant improvement. Nevertheless, we remain uncertain about the validity of the overall conclusion, that ketamine induces NMDA-R hypo-function in excitatory neurons, and that this effect is behaviorally manifested as an increase in a pro-variance bias.

Thank you for the summary of our revisions, and the opportunity to incorporate this new round of feedback. We believe these revisions, which include new figures and text, address the reviewers’ concerns and improve the manuscript through increased clarity. Importantly, we believe we have provided strong evidence to further support our main conclusion that ketamine induces NMDA-R hypofunction to lower E/I balance (by acting predominantly, but not necessarily exclusively, on excitatory neurons), and that this effect is behaviorally manifested as an increase in pro-variance bias.

Revisions for this paper:1) Motivate modeling approach.Given that you opted not to fit the model (which would be done with the meanfield reduction), or tune its parameters so that it matches the above behavioral patterns, we believe you should unpack the reasoning underlying this particular modeling approach.

We agree that further text would help to explain the reasoning behind our modeling approach.

We did not include direct fitting of the psychophysical data with circuit models for several reasons:

- We are not aware of any prior literature which has quantitively fit this class of circuit model – for either the spiking model or the mean-field reduction – directly to psychophysical behavior. We believe that developing approaches to do so is an important methodological challenge, but that it is beyond the scope of the present paper.

-Simulation of the spiking circuit model is too computationally expensive for model fitting.

- Fitting via the mean-field model reduction is a potentially tractable strategy. However, there are issues with the mean-field model related to its reduction which make it less than ideal. In particular, the effective noise parameter is added back, by hand as a free parameter, after the reduction. As such, this mean-field model does not derive what the magnitude of that noise parameter should be, nor how the strength of effective noise changes under a parameter perturbation. For this reason we do not use the mean-field model to examine E/I perturbations, as there is not a way to derive how the effective noise should vary across E/I perturbations which we expect would be important. (Instead, we used the mean-field model to examine the circuit mechanisms of the PVB phenomenon for a generic circuit.)

- Both spiking and mean-field models have a large number of parameters. It is not clear which parameters should be free and fitted vs. fixed. Even toward the conservative end, the number of plausibly fittable parameters is well over 10. Numerical simulation of the mean-field model needed for model fitting is still too computationally expensive in such a high-dimensional parameter space. There is not a principled reason to only fit over 2 dimensions. (In contrast, we did a 2D sweep over the NMDAR conductances, motivated by ketamine as a perturbation, to characterize their impact.)

- The parameterization of the mean-field model is not amenable to model fitting. Within the large number of parameters, there is a high degree of degeneracy, or “sloppiness” in how a parameter impacts psychophysical behavior, and parameters can effectively trade off each other at least locally. This is because the model is parameterized for biophysical mechanism rather than parameter parsimony at the level of behavioral output. This poses important – and largely unexplored – challenges for parameter identifiability and estimation, which are beyond the scope of the current study. Given these challenges, even if a model could be fit in the high-dimensional parameter space, it would be unclear how to interpret the set of fitted parameter values in light of potential degeneracies and how they may map onto lower-dimensional effective parameters (e.g., related to E/I ratio).

Although not well suited for model fitting to empirical behavioral data, biophysically-based circuit modeling can be fruitfully applied and interpreted for at least two purposes, which is why we chose this approach for this particular study:

- To examine whether, and through what dynamical circuit mechanism, a behavioral phenomenon (here, pro-variance bias) can emerge in biophysical circuit models within a particular dynamical regime (here, one previously developed to study decision making).

- To characterize how modulation of a biophysical parameter (here, NMDAR conductance, motivated by the pharmacological actions of ketamine) changes an emergent phenomenon (here, choice behavior) within a dynamical circuit regime.

Circuit modeling can demonstrate that a set of mechanisms is sufficient to produce a phenomenon. Furthermore, the pharmacological component of our study with ketamine naturally raises the question of how NMDAR hypofunction within this influential circuit model of decision making (Wang, 2002) impacts the behavioral phenomena studied here, which we examined from a bottom-up approach. Such a bottom-up approach is complementary to more top-down approaches of fitting behavior with computational- and algorithmic-level models. We believe that this modeling approach thereby provides useful insights even without behavioral model fitting, and furthermore it generates circuit-level predictions which can be investigated in future studies through experimental methods including electrophysiology and brain perturbations.

We have now added the following paragraph to the Discussion to note these issues with model fitting and the reasoning underlying our modeling approach:

“In this study we did not undertake quantitative fitting of the circuit model parameters to match the empirical data. […] The bottom-up mechanistic approach in this study, which makes links to the physiological effects of pharmacology and makes testable predictions for neural recordings and perturbations, is complementary to top-down algorithmic modeling approaches.”2) Plot model predictions along with data.As we pointed in our first review there seem to be some discrepancies between the data and the model, which we remain concerned about:i) Ketamine data asymptote at a lower than 100% level. The lapse rates are still not plugged in the circuit model so as to bring the model predictions closer to the data.

We thank the reviewers for bringing up these issues. For clarity, here lapse rate is defined as asymptote error rate at strong evidence. (Note that lapse rate is measured in completed trials, and therefore does not reflect uncompleted trials.) First, we would like to clarify our methods in the previous revision (previous Figure 8—figure supplements 4 and 5), which perhaps did not clearly emphasize how it accounted for lapse. The model comparisons to monkey data did indeed account for lapse rates for each monkey. Specifically, these analyses were comparing regression β weights between models and monkeys. Preceding that comparison, the regression β weights for the monkeys were calculated from a regression model that includes a lapse rate term (see Equations 8-11). Therefore, the model β weights were compared to lapse-corrected monkey β weights.

We see that for greater clarity it would be beneficial to visualize the model data that includes lapse rates at empirically-set levels, to facilitate direct comparison to the ketamine data. We have also decided to combine our response to this point with the related suggestion from comment 4 below to visualize the circuit model with empirical lapse rate, where we added a lapse mechanism downstream of the spiking circuit model. Specifically, we select a random subset of trials in the model, at a proportion according to the Monkey’s empirical lapse rate, and then randomize responses for those trials.

Finally, we have chosen to maintain Figure 7 without empirically-set lapses. We believe this is most logically consistent, with Figure 7 appearing chronologically first in the paper as a model prediction based on non-drug results, before lapses are demonstrated in ketamine data in Figure 8. Instead, the spiking circuit models with added empirically-set lapse rates are demonstrated in new supplementary figures (new Figure 8—figure supplements 8-9), for direct visual comparison to empirical ketamine results. In addition, we have added the empirically derived lapse rates to the results already presented in Figure 7—figure supplement 1.

ii) The control kernel in Figure 7I and the monkey kernels in Figure 8C look different. In the model, there is a primacy pattern (except for the first item) but in the data we see a flat/ U-shaped pattern. Plotting those together could reveal the degree of discrepancy.

Following the reviewers’ suggestion, we now include the juxtaposition of model and empirical plots, for the ketamine data, as new supplementary figures (new Figure 8—figure supplement 8-9; please see the prior comment above for details).

We also want to emphasize that the comparison of temporal weights might be more informative between control model (Figure 7I) and the non-drug data (Figures 2C,D), which both show a primacy effect. It is also interesting, and potentially important, to note that although the kernels differ somewhat between the control data in Figure 2, and the saline data in Figure 8 – namely, between showing more primacy vs. flat/U-shaped – both datasets show similar and robust pro-variance bias, which suggests that the precise shape of the kernel is not determinative of the pro-variance bias phenomenon.

The saline data (Figure 8G, Figure 8—figure supplement 1) might demonstrate a flat/ U-shaped pattern, distinct from both model and non-drug experimental data, for other reasons. For instance, the task structure for the saline/ketamine trials is different from that of the non-drug (and model) trials, with 6 instead of 8 stimuli, and was also made easier to keep the monkeys motivated (Please see “Task Modifications for Pharmacological Sessions” in Materials and methods for details). The difference between Figures 2C,D and Figure 8G might instead be due in part to such task modifications. Furthermore, we note that Figure 2 is based on about 7 times more trials than the saline data in Figure 8 and should therefore be more reliable. On the other hand, it is also possible the U-shaped pattern in the saline data suggests dynamical regimes in circuit models distinct from that considered here.

Motivated by this comment, we have added the following text to the Results:

“Additional observations further supported the lowered E/I hypothesis for the effect of ketamine on monkey choice behaviour. […] This may be due to task modifications for the ketamine/saline experiments compared with the non-drug experiments, but could also potentially arise from distinct regimes of decision making attractor dynamics (e.g. see Ortega et al., 2020).”

iii) In the control condition, the psychometric functions in Figure 7B and in Figure 8B look different (for example in terms of convergence of the light and dark coloured lines). The elevated E/I plot in Figure 7D appears to be closer to the saline psychometric curve.Such discrepancies, if true, matter: if the "baseline" model does not capture behavior in the control condition well, we cannot be confident about the validity of the subsequent perturbations performed to emulate the ketamine effect. To allow for better assessing the match, we strongly encourage you to always plot model predictions with the data.Ideally, you would also assess the goodness of fit using maximum likelihood. (A certain parametrization could exhibit similarity with the data in terms of the logistic regression weights (PVB) but at the same time it can miss largely on capturing the psychometric function.) We believe this would be straightforward, given the simulations you have already performed, but leave the decision to you, whether to not to do this.We realize that this point was not explicitly raised in the previous round. Then, reviewers had asked for a quantification of the goodness of fit. The approach you chose (logistic regression) is specific to the PVB index (not applied to psychometric functions and kernels) and did not fully convince reviewers.

We thank the reviewer for raising these important issues. We agree it will be beneficial to add a more direct comparison of the model behavior to each of saline and ketamine data, in both visualization and quantitative measures, in parallel to those in the last revision (which compared the effect of perturbation).

In brief, in the last revision, we focused primarily on comparing the model to the monkey ketamine behavior in terms of characterizing the *change* in behavioral measures of interest: namely, the mean weight, SD weight, and PVB index. We believe this is especially of interest because the change under ketamine is also important for comparing the two monkeys: for instance, two subjects may have very different baseline psychophysical performance, yet show a very consistent *change* from that baseline under ketamine. The same perspective applies to comparing the model to the monkeys, by focusing on the similarity of their change in behavior under a perturbation.

Nonetheless, we agree it is also of interest to provide assessment of how well the models’ psychometric performance agrees with the monkeys’ performance in saline and ketamine conditions. We have thus added a new supplementary figure which computes the Kullback-Leibler (KL) divergence between the saline and ketamine data of both monkeys and models of various perturbations (new Figure 8—figure supplement 6, NB. Figure 8—figure supplement 6 from the previous submission has moved to Figure 8—figure supplement 7). We chose KL divergence, instead of likelihood, because it was a more robust measure that was less sensitive to extreme responses in the behavior where the model had negligible likelihood (e.g., an error at strong evidence).

We note that we do not approach this as model fitting (for reasons elaborated in the response to the reviewers’ first comment above), but rather as providing a quantitative measure of psychometric similarity.

In this new analysis, we demonstrated the saline data for each monkey is more similar to the control model (green symbol) than to the lowered E/I model (purple) (Figure 8—figure supplement 6C, F), consistent with the previous conclusion. Importantly, while the elevated E/I plot in Figure 7D may appear visually more similar to the saline plot (combined across monkeys) in Figure 8B (as the reviewers pointed out), quantitatively comparing the model data to the saline plot (separated between the 2 monkeys) using KL divergence shows the control model is more similar to the saline data (Figure 8—figure supplement 6C, F). Finally, we would like to reiterate that the key model comparison determining the perturbation parameters in Figure 7 is still done with the previous perturbed vs baseline comparison (Figure 8—figure supplement 4,5,7), which we believe is at least as critical as the comparison in Figure 8—figure supplement 6. We have also changed the text to mention KL divergence wherever model comparison is mentioned.

In the new text, we now include these KL divergence results, alongside the measures of change in behavioral features. We have also expanded the “Testing the versatility of model predictions” section of Materials and methods to include KL divergence.

3) Assess effects of concurrent NMDA-blockade on E and I neurons.You establish that E/I increase reduces the PVB index while E/I decrease has the opposite effect. However, you have not examined the effect of concurrent changes of NMDA-Rs of both, E and I cells, which we had suggested to do. Please comment on the fact that concurrent changes could mimic the effect of E-E reduction (Figure 7, suppl. 2: moving up diagonally the purple point would result in equivalent behavior). Unless there is strong support in favor of the selective NMDA change over the concurrent change (assessed via maximum likelihood), the conclusions should be reframed.

We would like to clarify that we examined the effect of concurrent changes of NMDA-Rs on both E and I cells (e.g. please see Figure 7—figure supplement 2-3, Figure 8—figure supplement 4-5, and the newly added Figure 8—figure supplement 7 where we explicitly computed the resulting E/I ratio across EandI cells perturbation conditions). These 2D sweep analyses illustrate that the net effect on E/I ratio is a key effective parameter, and that concurrent changes can cause the same effects illustrated by the ‘pure’ perturbations. We have also decided to more strongly emphasize the discussion of concurrent changes of NMDA-Rs of both E and I cells, with a focus on the net effect on E/I ratio which can arise from concurrent changes. We take care not to suggest our findings support such a pure perturbation acting on a single cell type, which is not required for there to be a preferential impact on a cell type (e.g. via differential NMDA-R subunits) yielding a net impact on E/I ratio. We further added the following text in the Results section to justify why our main figures primarily considered perturbation to either E or I cells, but not both:

“[…] Crucially, the effect of E/I and sensory perturbations on PVB index and regression coefficients were generally robust to the strength and pathway of perturbation (Figure 7—figure supplement 2, 3).

Disease and pharmacology-related perturbations likely concurrently alter multiple sites, for instance NMDA-Rs of both excitatory and inhibitory neurons. We thus parametrically induced NMDA-R hypofunction on both excitatory and inhibitory neurons in the circuit model. The net effect on E/I ratio depended on the relative perturbation strength to the two populations^27^. Stronger NMDA-R hypofunction on excitatory neurons lowered the E/I ratio, while stronger NMDA-R hypofunction on inhibitory neurons elevated the E/I ratio. Notably, proportional reduction to both pathways preserved E/I balance and did not lower the mean evidence regression coefficient (a proxy of performance) (Figure 7—figure supplement 2A). […]”

We have also added an additional clarification to the Abstract to make clear that our main conclusion is that ketamine induces NMDA-R hypofunction to lower E/I balance (by acting predominantly, but not necessarily exclusively, on excitatory neurons):

“[…] Ketamine yielded an increase in subjects' PVB, consistent with lowered cortical excitation/inhibition balance from NMDA-R hypofunction predominantly onto excitatory neurons.”

4) Add a lapse rate downstream from circuit model.You have now assessed lapse rates in your analysis, but reviewers remarked that you do not report the best-fitting lapse rates. This makes it impossible to judge just how much lapses contribute to the decrease in task performance in the initial period following ketamine injection (which is included in all analyses). We are concerned that this massive performance drop under ketamine is not only triggered by aPVB, but also (perhaps largely) by an increase in lapses and a decrease in evidence sensitivity.We would expect a lapse mechanism to be in play in the circuit model when emulating the ketamine effect. You could use the fraction of lapses best fitted to psychometric curves (which clearly do not saturate at p(correct) = 1) for the circuit model simulations. It seems conceivable that allowing the circuit model to lapse will reduce the weight applied on the mean evidence.

We thank the reviewers for this excellent suggestion. We have added two supplementary figures, in which we incorporated a ‘downstream’ lapse mechanism to the circuit model, with lapse rate fitted to the two monkeys’ ketamine data (Figure 8—figure supplement 8-9). We believe this allows readers to better evaluate the effect of lapse using the circuit models.

We would also like to clarify that we have reported our best-fitted lapse rates in the previous submission. As shown in the previous submission, accounting for such lapse rates did not significantly change the regression weight or evidence sensitivities when analyzing the subjects’ behavior (Figure 8—figure supplement 2). Our further analyses have also shown the regression weights and evidence sensitivities are not affected in the circuit models’ behavior when empirical lapses are incorporated (Figure 8—figure supplement 8-9, please also see the new Figure 7—figure supplement 1). For clarity, we have further expanded on explaining the lapse rate.

“In further analysis, we also controlled for the influence of ketamine on the subjects’ lapse rate – i.e. the propensity for the animals to respond randomly regardless of trial difficulty. […] This confirmed that the rise in PVB was an accurate description of a common behavioral deficit throughout the duration of ketamine administration.”

Figure 8—figure supplements 8 and 9 can be found above in the response to comment 2. We also added the following text together with Figure 8—figure supplements 8 and 9:

“To quantify the effect of lapse rate on evidence sensitivity and regression weights in general, we examined the effect of a lapse mechanism downstream of spiking circuit models (Figure 8—figure supplement 8-9). Using the lapse rate fitted to the experimental data collected from the two monkeys, we assigned such portions of trials to have randomly selected choices for each circuit model, and repeated the analysis to obtain psychometric functions and various regression weights. Crucially, while the psychometric function as well as evidence mean and standard deviation regression weights were suppressed, the findings on PVB index were not qualitatively altered in the circuit models, further supporting the finding that the lapse rate does not account for changes in PVB under ketamine.”

5) Different quantification of pro-variance bias.We do not understand the motivation for compressing sensitivity to mean and to variance into a single PVB index. Our reading is that the pro-variance effect, quantified as a higher probability of choosing a more variable stream (see Tsetsos et al., 2012), can just be directly mapped onto the variance regressor. Combining the weights into a PVB index and framing the general discussion around this index seems unnecessary. The main behavioral result of ketamine can be parsimoniously summarized as a reduced sensitivity to the mean evidence. Relatedly, please discuss if and how the ketamine-induced increase in the PVB effect, the way you quantified it, rides over a strong decrease of the sensitivity to mean evidence under ketamine.It does seem to be the case that sensitivity to variance remains statistically indistinguishable between saline and ketamine (if anything it is slightly reduced). The E/I increase model consistently predicts that the variance regressor is reduced. This is not the case with the E/I decrease model, which occasionally predicts increases in the sensitivity to the variance (see yellow grids in Figure 7—figure supplement 2). This feature of the E/I decrease model should be discussed, as it seems to undermine the statement that the E/I perturbation produces robust predictions regardless of perturbation magnitude (i.e. depending on the strength of E/I reduction the model can produce a decrease or increase on variance sensitivity, and the relationship is non-monotonic). Overall, we believe that combining sensitivity to mean and variance obscures the interpretation of the data and model predictions.Again, we realize that this point appears to be new. But reviewers feel they could not really have a strong case regarding this metric without seeing the more detailed model predictions (in a 2-d grid) that you have presented in your revision.

We thank the reviewer for raising this point. We agree that further explanation of our choice to define a PVB measure as the ratio would improve the paper. We also agree that it is important to clearly report the effects of the mean and variance terms separately as well, to be explicit about what is driving the change in the ratio measure of PVB index.

First, as the reviewers note, there is a downside to reporting a ratio, which is that it can obscure how a change in the ratio is driven by changes in the numerator (SD) or denominator (mean) terms. Therefore, to accommodate the reviewers’ suggestion, we believe that the best solution for clarity is to report the changes in SD and mean weights, individually, alongside where changes in the ratio PVB index are reported. We have now included this information throughout the text, wherever a change in PVB index is reported for the model or monkeys, so that readers can readily keep track of how mean and SD terms are impacted, alongside the PVB index.

We believe that describing a PVB index as the ratio of SD to mean weights, is conceptually useful when interpreting *changes* in these behavioral sensitivities (as here by ketamine).

A key motivation relates to a point raised by the reviewers in the previous round of review, which said: “You should stress that the model does not feature any explicit PVB, and that PVB emerges through sample-by-sample competition between the two streams.” We agree that PVB should be understood as an emergent phenomenon arising from the decision-making process.

In evidence accumulation models, it is a non-trivial problem how to possibly reduce the sensitivity to the mean of evidence without a proportional reduction in the sensitivity to the SD of evidence. The simplest way to reduce the mean sensitivity would be down-scaling of the incoming evidence strength, but this would presumably down-scale the SD sensitivity by the same factor. Indeed, this is what is demonstrated by our “upstream deficit” perturbation: mean weight reduced, and SD weight reduced by the same proportion, leaving the PVB index unchanged. This is therefore a useful feature of our definition of PVB index: a ‘reference’ proportional change of SD weight is the same as the proportional change of mean weight, which results in no change in PVB index.

All three of our circuit perturbations (lowered E/I, elevated E/I, upstream deficit) reduce the sensitivity to the mean, and therefore consideration of the mean evidence regressor is not sufficient to dissociate these three circuit perturbations in the model. The qualitative behavioral dissociation between the three is their impact on the PVB index: increased PVB index for lowered-E/I, decreased PVB index for elevated-E/I, and unchanged PVB index for upstream deficit. Therefore, the key question to dissociate these three circuit perturbations is how the SD weight changes *relative* to the mean weight, which is captured by their ratio.

The reviewers are correct in pointing out that elevating E/I ratio consistently predicts that the evidence standard deviation regressor is reduced, while lowering E/I ratio can predict an increase in the sensitivity to the variance (as shown in Figure 7—figure supplement 2). This is, in fact, a non-trivial property of the circuit model. The circuit model, in the strong recurrent regime, has decision making choice accuracy following an inverted-U shape as a function of E/I ratio (e.g. see Lam et al., 2017). The control model is in fact not at the peak of this inverted-U shape, but slightly to the side. Instead, the peak of the inverted-U shape occurs at a weakly lowered E/I circuit. Similar to decision making choice accuracy, the evidence standard deviation regression weight also follows an inverted-U shape as a function of E/I ratio (Figure 7—figure supplement 2B). In contrast to choice accuracy, the control model is even further to the side away from the peak of the inverted-U shape, and the peak occurs at an E/I ratio lower than that of the peak for choice accuracy (e.g. compare Figure 7—figure supplement 2A and B).

If we consider the distinct locations of the control model on the inverted-U shape for choice accuracy and evidence standard deviation regression weight we explained above, the effects of elevating and lowering E/I ratio on the mean evidence regressor, evidence standard deviation regressor, and PVB index are more clear and interpretable. Elevating E/I ratio always moves the model down the inverted-U shapes for both choice accuracy and standard deviation regression weight, resulting in a consistent effect regardless of the scale of perturbation. On the other hand, weakly lowering E/I ratio would drive the model up the inverted-U shape for standard deviation regression weight, while driving the model down the inverted-U shape for choice accuracy, leading to a decrease in mean evidence regression weight but increase in standard deviation weight. It is only if the perturbation is strong enough to push the model across the peak (of standard deviation regression weight), that lowering E/I ratio will result in a decrease in both mean and standard deviation regressor. Notably, in both regimes the PVB index increases. This is another reason we utilized PVB index: as a robust measure of E/I ratio while alleviating the readers of the detailed changes and mechanisms of mean and standard deviation regression weights. (As an aside for completeness, when considering how the mean and standard deviation regression weights could possibly move along their respective inverted-U curves, an even weaker perturbation of lowered E/I will drive the control model up the peak of both inverted-U shapes. However, the range is smaller than the perturbation strengths used in this study, and the slight increase in mean evidence regression weight, which is bounded by the peak for choice accuracy which is not much higher than the already nearby control model, will be dominated by the larger increase in standard-deviation weight, where the control model is further from the peak and is not nearly as bounded. Therefore, this remains consistent with our argument that E/I perturbations produce robust effects regardless of perturbation magnitude).

We note that in our two monkey subjects, both showed strong significant decreases in mean weight, while only one showed a significant decrease in SD weight, yet both showed a consistent proportional change in PVB index. This is consistent, and can be explained by, the aforementioned inverted-U description.

Finally, another attractive property of presenting the PVB index as a ratio is that it is a dimensionless quantity, which facilitates comparisons between the monkeys and the model.

In the new revision, we have added in text noting changes in mean and SD weights where changes in PVB index are noted. We have also expanded the text when introducing the SD/mean ratio as a PVB index, to better motivate why it is an interesting measure for this phenomenon:

“In addition, we defined the pro-variance bias (PVB) index as the ratio of the regression coefficient for evidence standard deviation over the regression coefficient for mean evidence. […] From the ‘Regular’ trials, the PVB index across both monkeys was 0.173 (Monkey A = 0.230; Monkey H = 0.138).”

We have also added the following text in the Results to explain how the inverted-U phenomena relates to PVB index:

“Since the decision making choice accuracy depends on E/I ratio along an inverted-U shape – where the control, E/I balanced model is right next to the (slightly lowered E/I) peak (Lam et al., 2017) – both elevating and lowering E/I ratio drive the model away from the peak, resulting in lowered mean evidence regression weight. […]Notably, regardless of the magnitude with which E/I ratio is lowered, PVB index is consistently increased, providing a robust measure of pro-variance bias.”

We have also added the following text in Discussion to provide further motivation of the PVB index as a useful measure:

“The PVB index, as the ratio of standard deviation to mean evidence regression weights, serves as a conceptually useful measure to interpret changes in pro-variance bias due to ketamine perturbation in this study. […] The two monkeys, both interpreted as lowered E/I ratio using the model-based approach in this study, may therefore experience slightly different degrees of E/I reduction when administered with ketamine, as shown through concurrent changes in NMDA-R conductances in the circuit model (Figure 7—figure supplement 2).”